# Leave-One-Out Distinguishability in Machine Learning

**Jiayuan Ye[†], Anastasia Borovykh[‡], Soufiane Hayou[†], and Reza Shokri[†] [*]**
[†] National University of Singapore, [‡] Imperial College London

## Abstract

We introduce an analytical framework to quantify the changes in a machine learning algorithm's output distribution following the inclusion of a few data points in its training set, a notion we define as leave-one-out distinguishability (LOOD). This is key to measuring data **memorization** and information **leakage** as well as the **influence** of training data points in machine learning. We illustrate how our method broadens and refines existing empirical measures of memorization and privacy risks associated with training data. We use Gaussian processes to model the randomness of machine learning algorithms, and validate LOOD with extensive empirical analysis of leakage using membership inference attacks. Our analytical framework enables us to investigate the causes of leakage and where the leakage is high. For example, we analyze the influence of activation functions, on data memorization. Additionally, our method allows us to identify queries that disclose the most information about the training data in the leave-one-out setting. We illustrate how optimal queries can be used for accurate **reconstruction** of training data.[1]

## 1 Introduction

A key question in interpreting a model involves identifying which members of the training set have *influenced* the model's predictions for a particular query (Koh & Liang, 2017; Koh et al., 2019; Pruthi et al., 2020). Our goal is to understand the impact of the presence of a specific data point in the training set of a model on its predictions. This becomes important, for example, when model's prediction on a data point is primarily due to its own presence (or that of points similar to it) in the training set - a phenomenon referred to as *memorization* (Feldman, 2020) and can cause leakage of sensitive information about the training data. This enables an adversary to infer which data points were included in the training set (using *membership inference attacks*), by observing the model's predictions (Shokri et al., 2017; Zarifzadeh et al., 2023). One way to measure such **influence, memorization, and information leakage**, and thus addressing related questions, is by (re)training models with various combinations of data points. This empirical approach, however, is computationally expensive (Feldman & Zhang, 2020), and does not enable efficient analysis of, for example, what data points are influential, which queries they influence the most, and what properties of model (architectures) can impact their influence. These limitations demand an efficient and constructive modeling approach to influence and leakage analysis. This is the focus of the current paper.

We generalize the above-mentioned closely-related concepts (Section 2), and unify them as the *statistical divergence between the output distribution of models*, trained using (stochastic) machine learning algorithms, *when one or more data points are added to (or removed from) the training set*. We refer to this measure as leave-one-out distinguishability (LOOD). A larger LOOD of a training data $S$ in relation to query data $Q$ suggests a larger potential for information leakage about $S$, when an adversary observes the prediction of models trained on $S$ and queried at $Q$. As a special variant of LOOD, if we measure the gap between the *mean* of prediction distributions over leave-one-out models on a query, the resultant *mean distance LOOD* recovers existing definitions for influence (function) (Steinhardt et al., 2017; Koh & Liang, 2017; Koh et al., 2019) and memorization (self-influence) (Feldman & Zhang, 2020; Feldman, 2020) (i.e., a greater mean distance LOOD suggests a larger influence of training data $S$ on query data $Q$).

---

[*] Authors AB, SH and RS are ordered alphabetically.
[1] The code is available through this link.

We propose an *analytical method* that estimates LOOD accurately and efficiently (Section 3), by using Gaussian Processes (GPs) to model the output distribution of neural networks in response to querying on any data point(s).[2] We empirically validate that LOOD under NNGP models correlates well with the performance of membership inference attacks (Ye et al., 2022; Carlini et al., 2023) against neural networks in the leave-one-out setting for benchmark datasets (Figure 1b). We also experimentally show that mean distance LOOD under NNGP agrees with the prediction differences under leave-one-out retraining (Figure 1c)of deep neural networks. This demonstrates that our approach allows an accurate assessment of *information leakage and memorization (self-influence)* for deep neural networks, while significantly reducing the required computation time (by more than two orders of magnitude - Footnote 6) as we compute LOOD analytically.

Our method offers theoretical explanations as well as efficient algorithms (Section 4) to examine which query points are most influenced by any training data $S$ (i.e., to quantify the leakage of models in black-box settings). This further allows one to gauge the severity of this leakage by examining to what extent the most-influenced query can help **reconstruct the training data**. We prove that, for GPs under weak regularity conditions (Theorem 4.1), the differing data itself is a stationary point of LOOD. Experiments on image datasets further show that, LOOD under NNGP model is maximized when the query equals the differing datapoint (4b). These conclusions (for NNGP models) experimentally generalize to deep neural networks trained via SGD, where the empirical leakage is maximized when query data is within a small perturbation of the differing data (Figure 4c). Additionally, LOOD under NNGP remains almost unchanged even as the number of queries increases, even when the queries are optimized to have high LOOD (Appendix E.1). In other words, for NNGP models, all the information related to the differing point in the leave-one-out setting is contained within the model's prediction distribution solely on that differing point. We show how to exploit this to reconstruct training data in the leave-one-out or leave-one group-out settings. We present samples of reconstructed images in Figure 10 and 15-19, which strikingly resemble the training data, even when our optimization lands on sub-optimal queries.

Finally, the analytical structure of LOOD guides us in theoretically understanding how the model architecture can affect information leakage. In Section 6, we investigate how the analytical LOOD for Neural Network Gaussian Processes (NNGPs) is affected by activation functions. We prove that a low-rank kernel matrix implies low LOOD (Proposition 6.1) while existing literature (Proposition G.1) prove that the NNGP kernel for fully connected networks is closer to a low-rank matrix under ReLU activation (thus lead to less leakage) compared to smooth activation functions such as GeLU. Thus, smooth activation functions induce higher LOOD (information leakage) than non-smooth activation functions in deep neural networks. We validate that this conclusion aligns with empirical experiments for both NNGPs and for deep neural network trained via SGD (Figure 5).

## 2 LOOD DEFINITION AND ITS CONNECTIONS TO RELATED CONCEPTS

**Definition 2.1** (LOOD). Let $D$ and $D'$ be two training datasets that only differ in the set $S$.[3] Let $f_D(Q)$ be the distribution of model predictions given training dataset $D$ and query dataset $Q$, where the probability is taken over the randomness of the training algorithm. We define leave-one-out distinguishability (LOOD) as the statistical distance between $f_D(Q)$ and $f_{D'}(Q)$:

$$LOOD(Q; D, D') \coloneqq \text{dist}(f_D(Q), f_{D'}(Q)) \tag{1}$$

In the paper, we mainly measure LOOD using the KL divergence Kullback & Leibler (1951) to capture information leakage. However, as we explain below, different choices of $dist()$ and $f()$ allow us to recover established definitions of memorization, information leakage, and influence.

***Information Leakage as LOOD***. *Information Leakage* refers to how much a model's predictions can reveal information about specific data records in its training set, which would not be extractable if those entries were removed from the training set. In the black-box setting, where adversary observes model's prediction (distribution) at a query set $Q$, this is exactly LOOD. This is closely related

---

[2]NNGPs are widely used in the literature as standalone models (Neal, 2012), as modeling tools for different types of neural networks (Lee et al., 2017; Novak et al., 2018; Hron et al., 2020; Yang, 2019), and as building blocks for model architecture search (Park et al., 2020) or data distillation (Loo et al., 2022).

[3]$S = D \setminus D'$ and $D' \subset D$.

to Differential privacy (DP) (Dwork et al., 2006) — LOOD measures privacy loss in a black-box setting, for specific datasets, and DP determines the maximum (white-box) privacy loss across all possible datasets. [4] In this paper, we measure LOOD using KL divergence, which is different from Hocky-stick divergence used for equivalent definitions of $(\varepsilon, \delta)$-DP (Balle et al., 2018). The closest DP definition to LOOD is Rényi DP (Mironov, 2017) (consider Rényi divergence with order $\alpha \to 1$).

*LOOD gives useful information about leakage in DP ML algorithms.* DP guarantees that the prediction distributions cannot change beyond a certain bound under the change of any training data. However, such a guarantee reflects the worst-case scenario and the same bound holds for any possible training data and query. DP bound does not help measure the privacy risk of *specific* data points in certain training sets and queries. LOOD, however, helps compute this risk and explore how information leakage varies with different predictions, which prediction leads to the most leakage, and how leakage evolves with more predictions (See Figure 4, Section 4, and Appendix E.1).

*LOOD controls the leakage quantified by inference attacks.* One established method for empirically assessing information leakage is through the performance of a *membership inference attack* (MIA) (Shokri et al., 2017), where an attacker attempts to guess whether a specific data point $S$ was part of the model's training dataset. The power of any MIA (true positive rate given a fixed false positive rate) is controlled by the likelihood ratio test of observing the prediction given $D$ versus $D'$ as the training set (Neyman & Pearson, 1933). Thus, the leakage as quantified by MIA is upper bounded by statistical distance relates to likelihood ratio, e.g., KL divergence (Dong et al., 2019). [5] See Figure 1a for the high correlation between LOOD and MIA performance, and see Appendix C for why KL divergence is better suited than mean distance for measuring information leakage.

***Influence (Function) as (Derivatives of) Mean Distance LOOD***. The problem of influence estimation arises in the context of explaining machine learning predictions (Koh & Liang, 2017) and analyzing the robustness of a training algorithm against adversarial data (Steinhardt et al., 2017), where the influence of an (adversarial) training data $z \in D$ on the trained model's loss at test data $z_{test}$ is defined as $I_{loss}(z, z_{test}) = L(\hat{\theta}_{-z}; z_{test}) - L(\hat{\theta}; z_{test})$, where $\hat{\theta} = \arg\min L(\theta; D)$ on dataset $D$ and $\hat{\theta}_{-z} = \arg\min_\theta L(\theta; D \setminus \{z\})$. This quantity is thus the change of a trained model's loss on the test point $z_{test}$ due to the removal of one point $z$ from the training dataset (when $z = z_{test}$, it reflects self-influence of $z$). This is exactly LOOD given by mean distance (Definition C.1) between loss distributions, for an (idealized) training algorithm that outputs the optimal model given the training dataset. Estimating influence incurs computational difficulties, as it requires leave-one-out retraining and exact minimizers. Koh & Liang (2017) approximately compute this influence *without retraining* for *small perturbation* of training data, via a first-order Taylor approximation. They compute $I_{pert,loss}(z, z_{test}) = \nabla_\delta L(\hat{\theta}_{z_\delta, -z}; z_{test})$ where $\hat{\theta}_{z_\delta, -z} = \arg\min_\theta L(\theta; D) - L(\theta; z) + L(\theta; z_\delta)$ via non-exact estimations of the Hessian (as exact Hessian computation is computationally expensive for large model). For simple models such as linear model and shallow CNN, the influence function approximation correlate well with leave-one-out retraining. However, for deep neural networks, the influence function approximation worsens significantly ((Koh & Liang, 2017, Figure 2) and Figure 1c). By contrast, we propose an analytical method for computing mean distance LOOD that shows high correlation to leave-one-out retraining even for deep neural networks (Figure 1c).

***Memorization as LOOD***. Memorization, as defined in (Feldman, 2020, page 5), is given by $mem(A, D, S) = Pr_{h \sim A(D)}(h(x_S) = y_S) - Pr_{h \sim A(D \setminus S)}(h(x_S) = y_S)$; this measures the change of prediction probability on training data $S$ after removing it from the training dataset $D$, given training algorithm $A$. To avoid training many models on $D$ and $D \setminus S$ for each pair of training data and dataset, one approach is to resort to boosting-like techniques (Feldman & Zhang, 2020; Zhang et al., 2021) to reduce the number of retrained models (at a drop of estimation quality), which still requires training hundreds or thousands of models. The memorization measure is exactly mean distance LOOD Definition C.1 when query equals the differing data (i.e., self-influence) and output function $f$ is the prediction accuracy. Thus, we can analytically compute and reason about memorization in ML using LOOD.

---

[4]DP guarantee certainly applies to black-box setting too, but its primary analysis is done for the stronger white-box threat model, where adversary observes the model parameters (or parameter updates during training).

[5]KL divergence (and thus LOOD) can also be used for semantically measuring the advantage of reconstruction attacks and attribute inference attacks. See (Guo et al., 2022, Section 5.2) for a concrete example.

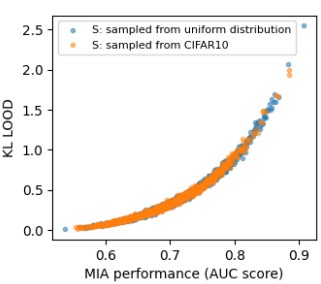

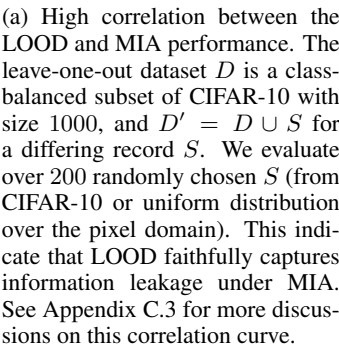

(a) High correlation between the LOOD and MIA performance. The leave-one-out dataset $D$ is a class-balanced subset of CIFAR-10 with size 1000, and $D' = D \cup S$ for a differing record $S$. We evaluate over 200 randomly chosen $S$ (from CIFAR-10 or uniform distribution over the pixel domain). This indicate that LOOD faithfully captures information leakage under MIA. See Appendix C.3 for more discussions on this correlation curve.

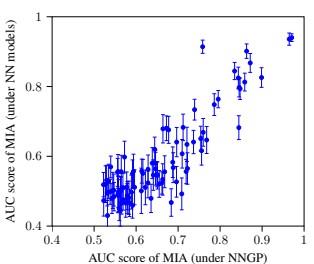

(b) MIA performance on NNGP and NN models matches, for 90 random training data records, each represented by a dot. For MIA on NNs, we evaluate over 50 models on $D$ and 50 models on $D'$. For MIA on NNGP, we compute the prediction mean and variance of NNGP (on the differing data) given $D$ and $D'$, and evaluate MIA over 5000 samples from the two Gaussian distributions. Models are trained on 'car' and 'plane' images from CIFAR10.

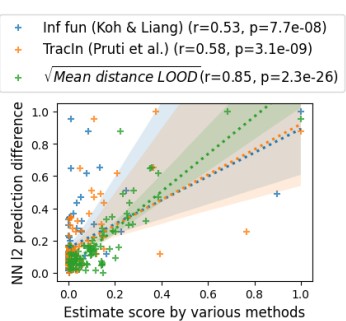

(c) Mean distance LOOD matches the prediction difference after leave-one-out retraining of NN models, for 90 randomly chosen differing data (each represented by a dot). Dataset and model setups are the same as Figure 1b. As comparison, we plot the influence function estimates (Koh and Liang [10]) and TracIn scores (Pruthi et al.[18]). We normalize all estimates to $[0, 1]$. We report Pearson's correlation and p-value in the legends.

Figure 1: Validation of our analytical framework (based on LOOD and mean distance LOOD for NNGP), according to information leakage measure (the performance of membership inference attacks) and influence definition (prediction difference under leave-one-out retraining).

## 3 OUR ANALYTICAL FRAMEWORK FOR ESTIMATING LOOD

Prior approaches for estimating leakage and memorization are heavily based on experiments, thus suffering from (a) approximation error (e.g., first-order Taylor expansion and non-exact Hessian approximation); (b) modelling error (e.g., suboptimal MIAs for quantifying leakage); (c) high computation cost (e.g., training many models for accurate estimation of memorization); and (d) experimental instability. To address this, we propose an *analytical* method that estimates LOOD *accurately* and *efficiently*, by modelling the prediction distribution of neural networks as *Gaussian process*.

***Gaussian process***.    To analytically capture the randomness of machine learning on model outputs we use Gaussian Process (GP). A GP $f \sim GP(\mu, K)$ is specified by its mean function $\mu(\mathbf{x}) = \mathbb{E}_f[f(\mathbf{x})]$ and covariance kernel function $K(\mathbf{x}, \mathbf{x}') = \mathbb{E}_f[(f(\mathbf{x}) - \mu(\mathbf{x}))(f(\mathbf{x}') - \mu(\mathbf{x}'))]$ for any inputs $\mathbf{x}$ and $\mathbf{x}'$ (Williams & Rasmussen, 2006). The mean function is typically set to zero, under which the kernel function $K$ completely captures the structure of a GP. Some commonly used kernel functions are isotropic kernels and correlation kernels (Appendix D.2). Given training data $D$ with noisy labels $y_D = f(D) + \mathcal{N}(0, \sigma^2\mathbb{I})$, the posterior prediction distribution of GP on a query data $Q$ follows multivariate Gaussian $f_{D,\sigma^2}(Q) \sim \mathcal{N}(\mu_{D,\sigma^2}(Q), \Sigma_{D,\sigma^2}(Q))$ with analytical mean $\mu_{D,\sigma^2}(Q) = K_{QD}(K_{DD} + \sigma^2\mathbb{I})^{-1}y_D$ and covariance matrix $\Sigma_{D,\sigma^2}(Q) = K_{QQ} - K_{QD}(K_{DD} + \sigma^2\mathbb{I})^{-1}K_{DQ}$ (by GP regression). Here for brevity, we denote $K_{XZ} = (K(\mathbf{x}_i, \mathbf{z_j}))_{x_i \in D, z_j \in Z}$ as the kernel matrix between $X$ and $Z$.

***Neural Network Gaussian process and connections to NN training***.  Prior works (Lee et al., 2017; Schoenholz et al., 2017; 2016) have shown that the output of neural networks on a query converges to a Gaussian process, at initialization as the width of the NN tends to infinity. The kernel function of the neural network Gaussian process (NNGP) is given by $K(\mathbf{x}, \mathbf{x}') = \mathbb{E}_{\theta \sim \text{prior}}[\langle f_\theta(\mathbf{x}), f_\theta(\mathbf{x}')\rangle]$, where $\theta$ refers to the neural network weights (sampled from the *prior* distribution), and $f_\theta$ refers to the prediction function of the model with weights $\theta$. In this case, instead of the usual gradient-based training, we can analytically compute the Gaussian Process regression posterior distribution of the network prediction given the training data and query inputs. See Appendix B for more discussions on the connections between NNGP and the training of its corresponding neural networks.

***LOOD for NNGP effectively captures leakage in practical NN training***.  Under Gaussian process (GP) modelling, we can analytically compute and analyze LOOD on the GP models associated with a given neural network (i.e., NNGP). For the underlying NNGP models, we first investigate whether the LOOD accurately reflects the level of information leakage in the idealized leave-one-out membership inference attack (Ye et al., 2022, Section 4.5). Specifically, the leave-one-out MIA distinguish between two prediction distributions and of models trained on leave-one-out datasets that only differs in one record $S$. In Figure 1a, we indeed observe a strong correlation between LOOD and MIA performance. This concludes that LOOD effectively reflects information leakage in NNGP. We also empirically validate whether the leakage under NNGP models accurately reflects the leakage in neural networks trained using SGD. In Figure 1b, we observe a strong correlation between the MIA performance on NNGP models and the MIA performance on neural network models. This suggests that NNGP captures the leakage in the corresponding neural networks.

***Mean distance LOOD for NNGP agrees with memorization in practical NN training***.  We further investigates whether our analytical framework allows accurate estimates for memorization (self-influence). In Figure 1c, we show the estimates given by different methods ($y$-axis), and compare it with the actual prediction difference ($x$-axis) between 100 NN models trained on each of the leave-one-out datasets, over 50 randomly chosen differing data. That is, the $x$-axis is exactly the memorization or self-influence as defined by (Feldman, 2020, Page 5) and (Feldman & Zhang, 2020, Equation 1), and serves as a ground-truth baseline that should be matched by good influence estimations. In the $y$-axis, we show the normalized mean distance LOOD (under NNGP), influence function (Koh & Liang, 2017) and TracIn estimates (Pruthi et al., 2020). We observe that mean distance LOOD has the highest correlation to the actual prediction difference among all methods.

***Analytical LOOD allows improved computational efficiency***.   One significant advantage of our method is the efficiency of computing analytic expressions. In Figure 1b- 1c, we observed over $140\times$ **speed up** for estimating leakage and influence using our framework versus empirically measuring it over retrained models as it is performed in the literature (Ye et al., 2022; Feldman & Zhang, 2020).[6] This allows us to efficiently answer many advanced questions (that are previously computationally expensive or infeasible to answer via experimental approaches), such as: Where, i.e., at which query $Q$, does the model exhibit the most information leakage or influence of its training data? How do different model architectures impact influence? We discuss these questions in Section 4-6.

## 4 OPTIMIZING LOOD TO IDENTIFY QUERY WITH MAXIMUM LEAKAGE

LOOD (6) measures the information leakage of a model about its training data $S$ when queried on a query data point $Q$. For assessing the maximum possible leakage, we need to understand *which query $Q$ should the attacker use to extract the highest possible amount of information about $S$*. This question is closely related to the open problem of constructing a worst-case query data for tight privacy auditing via MIAs (Steinke et al., 2024; Jagielski et al., 2020; Nasr et al., 2021; Wen et al., 2022; Nasr et al., 2023). However, evaluating leakage via MIAs can be computationally expensive due to the associated costs of retraining many models on a fixed pair of leave-one-out datasets. To address this, we will show that estimating LOOD under GPs offers an efficient alternative for answering this question. In this section, we consider *single* query ($|Q| = 1$) and single differing point ($|S| = 1$) setting and analyze the most influenced point by maximizing the LOOD:

$$\arg\max_{Q} LOOD(Q; D, D') \coloneqq dist(f_{D,\sigma^2}(Q), f_{D',\sigma^2}(Q)), \qquad (2)$$

We theoretically analyze and empirical solve the LOOD optimization problem (2). Our analysis also extends to the multi-query ($|Q| > 1$) and a group of differing data ($|S| > 1$) setting (Appendix E).

***Analysis: querying the differing data itself incurs stationary LOOD (information leakage)***.  We first prove that for a wide class of kernel functions, the differing point is a stationary solution for (2), i.e., satisfies the first-order optimality condition. In the following theorem, we prove that if the kernel function satisfies certain regularity conditions, the gradient of LOOD at the differint point is zero, i.e., querying the differing incurs stationary LOOD over its local neighborhood.

---

[6]Evaluating the MIA and prediction difference under leave-one-out retraining in Figure 1b took 100 GPU hours for training $50 \times 90$ FC networks. For the same setting, estimating LOOD only took 40 GPU minutes.

**Theorem 4.1.** *Let the kernel function be $K$. Assume that (i) $K(x,x) = 1$ for all training data $x$, i.e., all training data are normalized under kernel function $K$; (ii) $\frac{\partial K(x,x')}{\partial x}\mid_{x'=x} = 0$ for all training data $x$, i.e., the kernel distance between two data records $x$ and $x'$ is stable when they are equal. Then, the differing data is a stationary point for LOOD as a function of query data, as follows*

$$\nabla_Q \text{LOOD}(f_{D,\sigma^2}(Q)\|f_{D',\sigma^2}(Q)) \mid_{Q=S} = 0,$$

*where by definition we have that* $\nabla_Q \text{LOOD}(f_{D,\sigma^2}(Q)\|f_{D',\sigma^2}(Q)) = \left(1 - \frac{\Sigma'(Q)}{\Sigma(Q)}\right) \cdot \nabla_Q \left(\frac{\Sigma(Q)}{\Sigma'(Q)}\right) + \nabla_Q \left(\|\mu(Q) - \mu'(Q)\|_2^2\right)\Sigma'(Q)^{-1} + \|\mu(Q) - \mu'(Q)\|_2^2 \nabla_Q(\Sigma'(Q)^{-1})$

The full proof is deferred in Appendix D.1. Theorem 4.1 proves that as long as the kernel function $K$ satisfies weak regularity conditions (which hold for commonly used RBF kernel and NNGP kernel on a sphere as shown in Appendix D.2), the first-order stationary condition of LOOD is satisfied when the query equals differing data. This stationarity is a necessary condition for maximal leakage, and suggests that the differing point itself could be the query that incurs maximal leakage.

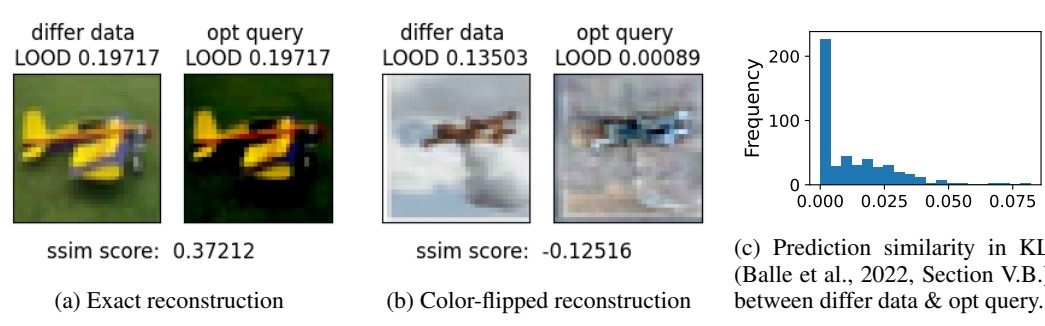

(a) Exact reconstruction      (b) Color-flipped reconstruction

(c) Prediction similarity in KL (Balle et al., 2022, Section V.B.) between differ data & opt query.

Figure 2: Examples of optimized query after LOOD optimization, given all 'car' and 'plane' images from CIFAR10 as training dataset, under NNGP for 10-layer FC network with ReLU activation. We show in (a) the optimized query gets close to the differing point in LOOD; and in (b) the optimized query converges to a query with significantly lower LOOD than the differing data, yet they visually resemble each other. In (c) we further statistically check the prediction similarity between query and differing data in KL divergence (Balle et al., 2022, Section V.B.) across 500 runs with randomly chosen differing data (see Figures 14, 15, 16, 17, 18 and 19 for all the images for these runs).

***Experimentally Optimizing LOOD enables a Data Reconstruction Attack.*** To further understand what query $Q$ induce maximal leakage about training data $S$, we run Adam on the non-convex objective (2). Interestingly, we observe that the *optimized query by* (2) *generally recovers the differing data with high visual precision* across random experiments: see Figure 2a, Appendix D.3 and Figure 14-19 for examples under RBF and NNGP kernels. [7] The optimized query also tends to have very similar LOOD to the differing data itself (see Figure 2c), even when there are more than one differing data, i.e., leave-one-group-out setting (see Figure 3 – only in its right-most column does the optimized single query have slightly higher LOOD than the differing record that it recovers). We remark that, possibly due to the nonconvex structure of LOOD, cases exist where the optimized query has significantly lower LOOD than the differing data (Figure 2b). Noteworthy is that even in this case the query still resembles the shape of the differing data. This shows the significant amount of information leaked through the model's prediction on the differing data itself. Such results show that LOOD can be leveraged for a *data reconstruction attack* in a setting where the adversary has access to the prediction distribution of models trained on leave-one-out datasets. This setting aligns with prior works (Balle et al., 2022; Guo et al., 2022) that focus on reconstruction attacks under the assumption that the adversary possesses knowledge of *all* data points except for one.

Our reconstruction via optimizing LOOD methodology extends to the group setting. Figure 3 demonstrates that even when the attacker does not know a *group* of data points, they can successfully

---

[7]We tried several image similarity metrics in the reconstruction literature, e.g., per-pixel distance (Carlini et al., 2023, Section 5.1), SSIM (Haim et al., 2022, Figure 1) and LPIPS (Balle et al., 2022, Section V.B) – but they do not reflect the visual similarity in our setting. This is possibly because LOOD optimization only recovers the highest amount of information about the differing data which need not be the exact RGB values. It is an open problem to design better image similarity metrics for such reconstructions beyond RGB values.

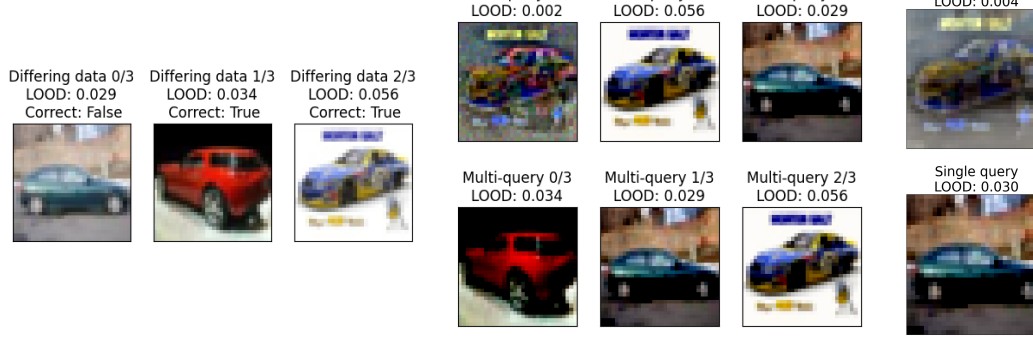

Figure 3: Query optimization (data reconstruction) by leave-one-group-out distinguishability, when there are three differing records (the three leftmost images). Above each differing record, we show whether it is predicted correctly by the NNGP model trained without it. The three middle images are the optimal queries in the multi-query setting, and the rightmost image is the optimal single query; each row presents the results for a different random query initialization. Observe that differing images with highest LOOD (and for certain initialization even *all* differing images) can be recovered.

reconstruct the whole group with high visual similarity. Additionally, by examining the frequency of LOOD optimization converging to a query close to each data point in the differing group (across optimized queries in multiple runs), we can assess which $S$'s are most vulnerable to reconstruction (Appendix E.2). Reconstruction under the leave-one-group-out setting mimics a practical real world adversary that aims to reconstruct more than one training data. In the limit of large group size that equals the size of the training dataset, such leave-one-group-out setting recovers one of the difficult reconstruction attack scenario studied in (Haim et al., 2022), where the adversary does not have exact knowledge about any training data. Therefore, extending our results within the leave-one-group-out setting to practical reconstruction attacks presents a promising open problem for future exploration.

***Experimental maximality of leakage when query equals differing data for NNGPs and NNs.*** To complement Theorem 4.1 and explain the success of our reconstruction attack, we experimentally investigate whether querying on differing point itself incurs maximal LOOD. We observe many empirical evidences that support this hypothesis, e.g., for a one-dimensional toy dataset in Figure 4a. In Figure 4b and its caption, we further observe that querying on differing data $S$ itself consistently incurs larger LOOD than querying on random perturbations of $S$ for CIFAR10 under NNGP. Appendix E also support this empirical maximality of LOOD under multiple queries and a group of differing data (Appendix E). Finally, Figure 4c and its caption show that MIA performance on NN models is generally the highest when queries are near the differing data. On the other hand, we note that in practice data-dependent normalization is often used prior to training. In such a setting there exists *optimized query with higher LOOD than the differing data itself* – see Appendix D.4.

## 5 OPTIMIZING LOOD TO IDENTIFY THE MOST INFLUENCED POINT

Mean distance LOOD (Definition C.1) measures the influence of training data $S$ on the prediction at query $Q$. Thus we can use mean distance LOOD to *efficiently* analyze *where (at which $Q$) the model is most influenced by the change of one train data $S$*. This question shed light on the fundamental question of which prediction is influenced the most by (adversarial) training data (Koh & Liang, 2017) (and from there connections to robustness against data poisoning Steinhardt et al. (2017)).

We now use mean distance LOOD for a more fine-grained analysis of influence and memorization within our framework. In Proposition 5.1, we analyze the different behaviors of mean distance LOOD, based on where the differing point is located with respect to the rest of the training set.

**Proposition 5.1** (First-order optimality condition for influence)**.** *Consider a dataset $D$, a differing point $S$, and a kernel function $K$ satisfying the same conditions of Theorem 4.1. Then, we have that*

$$\nabla_Q M(Q) \mid_{Q=S} = -\alpha^{-2}(1 - K_{SD}M_D^{-1}K_{DS})(y_S - K_{SD}M_D^{-1}y_D)^2 \mathring{K}_{SD}M_D^{-1}K_{DS},$$

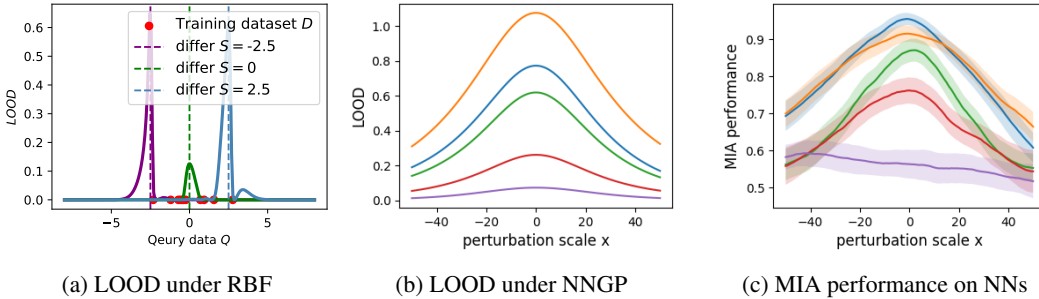

(a) LOOD under RBF  (b) LOOD under NNGP  (c) MIA performance on NNs

Figure 4: Empirical maximality of information leakage when query equals differing data. Figure (a) uses a toy dataset $D = (x, y)$ with $x \in [-5, 5]$, $y = \sin(x)$, and computes LOOD for all queries in the data domain. Figure (b), (c) use 'car' and 'plane' images in CIFAR-10, and evaluate leakage on perturbation queries $\{Q = S + x \cdot R : x \in [-50, 50]\}$ of the differing data $S$ along a random direction $R \in L_\infty$ unit ball. For five randomly chosen differing data, we show in (b) and (c) that LOOD and MIA performance are visually maximal around $x = 0$. For statistical pattern, we repeat the experiment for 70 randomly chosen differing data, and observe the scale of perturbation with maximal information leakage is $x = 0.000 \pm 0.000$ for NNGPs, and $x = -2.375 \pm 5.978$ for NNs.

*where $M(Q)$ is the mean distance LOOD in C.1, $K_{SD}, K_{DS}, K_{DD}$ are kernel matrices as defined in Section 2, $\mathring{K}_{SD} = \frac{\partial}{\partial Q} K_{QD} |_{Q=S}$, $M_D = K_{DD} + \sigma^2 I$, and $\alpha = 1 - K_{SD} M_D^{-1} K_{DS} + \sigma^2$.*

Proposition 5.1 proves that when $S$ is far from the dataset $D$ (i.e., $K_{SD} \approx 0$) or close to the dataset $D$ (i.e., $y_S \approx K_{SD} M_D^{-1} y_D$, that is $S$ is perfectly predicted by models trained on $D$), the mean distance LOOD gradient $\nabla_Q M(Q) |_{Q=S}$ is close to 0 (i.e., the differing data $S$ has stationary influence on itself). When $S$ is far from the remaining training dataset $D$ and the data labels are bounded (which is the case in a classification task), we further analyze the Hessian of mean distance LOOD and prove that differing data $S$ has locally maximal influence on itself (Lemma F.4). Between the two cases, when the differing data is neither too close nor too far from the remaining dataset, Proposition 5.1 proves that $\nabla_Q M(Q) |_{Q=S} \neq 0$, i.e., the differing data $S$ does not incur maximal influence on itself. To give an intuition, we can think of the Mean LOOD as the result of an interaction of two forces, a force due to the training points, and another force due to the differing point. When the differing point is neither too far nor too close to the training data, the synergy of the two forces creates a region that does not include the differing point where the LOOD is maximal. See Figures 11, 12 and 13 for more experimental illustrations of the behaviors of mean distance LOOD.

## 6 EXPLAINING THE EFFECT OF ACTIVATION FUNCTIONS ON LOOD

LOOD under NNGP enjoys analytical dependence on model architectures, thus enabling us to efficiently investigate questions such as: *how does the choice of activation function affect the magnitude of information leakage?* Answers to such questions would provide guidance for how to train models in a more privacy-preserving manner. To theoretically analyze how the choice of activation function affect the magnitude of LOOD, we will leverage Proposition G.1 in the Appendix; it shows that for the same depth, smooth activations such that GeLU are associated with kernels that are farther away from a low rank all-constant matrix (more expressive) than kernel obtained with non-smooth activations, e.g. ReLU. Therefore to understand the effect of activation, it suffices to analyze how the low-rank property of kernel matrix affects the magnitude of LOOD. Thus, we establish the following proposition, which shows that LOOD is small if the NNGP kernel matrix has low rank property.

**Proposition 6.1** (Informal: Low rank kernel matrix implies low LOOD). *Let $D$ and $D' = D \cup S$ be an arbitrarily fixed pair of leave-one-out datasets, where $D$ contains $n$ records. Define the function $h(\alpha) = \sup_{x,x' \in D'} |K(x, x') - \alpha|$. Let $\alpha_{min} = \text{argmin}_{\alpha \in \mathbb{R}^+} h(\alpha)$, and thus $h(\alpha_{min})$ quantifies how close the kernel matrix is to a low rank all-constant matrix. Assume that $\alpha_{min} > 0$. Then, there exist constants $A_n, B > 0$ such that*

$$\max_Q LOOD(f_{D,\sigma^2}(Q) \| f_{D',\sigma^2}(Q)) \leq A_n h(\alpha_{min}) + B n^{-1}.$$

*Thus the smaller $h(\alpha_{\min})$ is, the smaller LOOD could be. See G for formal statement and proof.*

Proposition 6.1 proves that the LOOD (leakage) over worst-case query is upper bounded by a term $h(\alpha_{min})$ that is small if the kernel matrix is close to a low-rank all-constant matrix. Intuitively, this happens because the network output *forgets* most of the information about the input dataset under low rank kernel matrix. By combining Proposition G.1 and Proposition 6.1, we conclude that *smooth activations induce higher leakage than non-smooth activations*. This is consistent with the intuition that smooth activations allow deeper information propagation, thus inducing more expressive kernels and more memorization of the training data. Interestingly, recent work (Dosovitskiy et al., 2021) also showed that GeLU enables better model performance than ReLU even in modern architectures such as ViT. Our results thus suggest there is a privacy-accuracy tradeoffs by activation choices.

***Experimental results on NNGP validates that GeLU activation induces higher LOOD than ReLU activation***. We now empirically investigate how the activation function affects the LOOD under NNGP. We evaluate NNGP for fully connected neural networks with varying depths $\{2, 4, \cdots, 10\}$, trained on 'plane' and 'car' images from CIFAR10. We compute LOOD over $500$ randomly chosen differing data, and observe under fixed depth, the LOOD under ReLU is more than $1.1\times$ higher than LOOD under GeLU in $95\%$ cases. We visualize randomly chosen 20 differing data in Figure 5. For each fixed differing data, we also observe that the gap between LOOD under ReLU and LOOD under GeLU grows with the depth. This is consistent with the pattern predicted by our analysis in Proposition G.1, i.e., GeLU gives rise to higher leakage in LOOD than ReLU activation, and their gap increases with depth. It is worth noting that our experiments for the LOOD optimization are done without constraining the query data norm. Therefore, our experiments shows that Proposition G.1 could be valid in more general settings.

***Experimental results on NNs validate the effect of activation choice on leakage***. We further investigate whether leakage (measured by MIA performance on leave-one-out retrained models) for deep neural networks are similarly af-

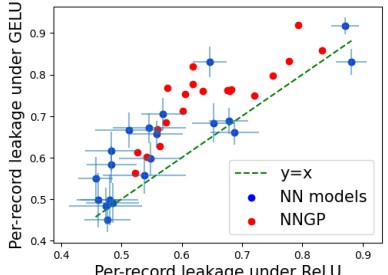

Figure 5: Empirical validation of our analytical results in of Section 6 that per-record information leakage is higher under GELU activation than under ReLU, for both NNGPs and NN models. We evaluate per-record leakage with membership inference attack performance on models trained on leave-one-out datasets. Dataset contains 'car' and 'plane' images from CIFAR-10. The NN model is fully connected network with depth 10 and width 1024.

fected by activation choice. Figure 5 shows that DNNs with GeLU activation incurs higher per-record leakage (MIA performance in AUC score) compared to DNNs with ReLU activation, for 20 randomly chosen differing data. This is consistent with our analytical results (Proposition G.1 and 6.1), and aligns with prior empirical observations Wei et al. (2020); Haim et al. (2022); Shamsabadi et al. (2023) that data reconstruction is harder under ReLU activation (than its smooth counterparts).

## 7 CONCLUSIONS

We show that the estimations of information leakage, influence, and memorization are intrinsically the same problem of estimating leave-one-out distinguishability (LOOD). We propose one analytical framework for accurately and efficiently computing LOOD in machine learning via Gaussian process modeling. This framework facilitates more efficient and interpretable answers to existing questions and enables us to explore new questions about how a data point influences model predictions. Notably, the analytical nature of LOOD enables performing optimization to identify predictions that leak the most information about each training data record (Section 4), which enables exact *reconstruction* of each training data.

***Future Works***. An interesting direction is to use LOOD to analyze how other factors affect information leakage, such as the training data (distribution) and architecture choices. It is also interesting to extend our framework for estimating other concepts closely related to leave-one-out distinguishability, such as Shapley value (Shapley et al., 1953) in the literature of data valuation. [8]

---

[8]Shapley value is a linear combination of LOOD on all subsets of the training dataset. Thus, in certain scenarios e.g., submodular utility, LOOD regarding the complete dataset is a lower bound for Shapley value.

ACKNOWLEDGEMENTS

The authors would like to thank Martin Strobel for the help in reproducing influence function, and to thank Yao Tong, Tuan-Duy H. Nguyen and Yaxi Hu for helpful feedback on earlier drafts of the paper. The work of Reza Shokri was supported by the Asian Young Scientist Fellowship 2023, and the Ministry of Education, Singapore, Academic Research Fund (AcRF) Tier 1.

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

CONTENTS

# A  TABLE OF NOTATIONS

Table 1: Symbol reference

| Symbol | Meaning |
|---|---|
| $D$ | Leave-one-out dataset |
| $D'$ | Leave-one-out dataset combined with the differing data |
| $S$ | Differing data record(s) |
| $s$ | number of differing data records |
| $Q$ | Queried data record (s) |
| $q$ | number of queries |
| $d$ | dimension of the data |
| $K(x, x')$ | Kernel function |
| $K(X, Z) \in \mathbb{R}^{p \times k}$, for two finite collections of vectors $X = (x_1, \ldots, x_p)$ and $Z = (z_1, \ldots, z_k)$ | $(K(x_i, z_j))_{\substack{1 \leq i \leq p \\ 1 \leq j \leq k}}$ |
| $K_{XZ} \in \mathbb{R}^{p \times k}$ | Abbreviation for $K(X, Z)$ |
| $M_D$ | Abbreviation for $K_{DD} + \sigma^2 \mathbb{I}$ |
| $M_{D'}$ | Abbreviation for $K_{D'D'} + \sigma^2 \mathbb{I}$ |
| $M(Q)$ | Abbreviation for mean distance LOOD under query $Q$ and leave-one-out datasets $D$ and $D'$ |
| MIA | Abbreviation for membership inference attack |

# B  ADDITIONAL DISCUSSION ON THE CONNECTIONS BETWEEN NNGP AND NN TRAINING

Note that in general, there is a performance gap between a trained NN and its equivalent NNGP model. However, they are fundamentally connected even when there is a performance gap: the NNGP describes the (geometric) information flow in a randomly initialized NN and therefore captures the early training stages of SGD-trained NNs. In a highly non-convex problem such as NN training, the initialization is crucial and its impact on different quantities (including performance) is well-documented Schoenholz et al. (2016). Moreover, it is also well-known that the bulk of feature learning occurs during the first few epochs of training (see (Lou et al., 2022, Figure 2)), which makes the initial training stages even more crucial. Hence, it should be expected that properties of NNGP transfer (at least partially) to trained NNs. More advanced tools such as Tensor Programs Yang & Littwin (2023) try to capture the covariance kernel during training, but such dynamics are generally intractable in closed-form.

# C  DEFERRED DETAILS FOR SECTION 2

## C.1  MEAN DISTANCE LOOD

The simplest way of quantifying the LOOD is by using the distance between the means of the two distributions formed by coupled Gaussian processes:

**Definition C.1** (Mean distance LOOD). By computing the distance between the mean of coupled Gaussian processes on query set $Q$, we obtain the following mean distance LOOD.

$$M(Q; D, D') = \frac{1}{2} \|\mu_{D,\sigma^2}(Q) - \mu_{D',\sigma^2}(Q)\|_2^2 \tag{3}$$

$$= \frac{1}{2} \|K_{QD}(K_{DD} + \sigma^2 \mathbb{I})^{-1} y_D - K_{QD'}(K_{D'D'} + \sigma^2)^{-1} y_{D'}\|_2^2. \tag{4}$$

where $Q$ is a query data record, and $\mu_{D,\sigma^2}(Q)$ and $\mu_{D',\sigma^2}(Q)$ are the mean of the prediction function for GP trained on $D$ and $D'$ respectively, as defined in Section 2. When the context is clear, we abbreviate $D$ and $D'$ in the notation and use $M(Q)$ under differing data $S$ to denote $M(Q; D, D')$ where $D' = D \cup S$.

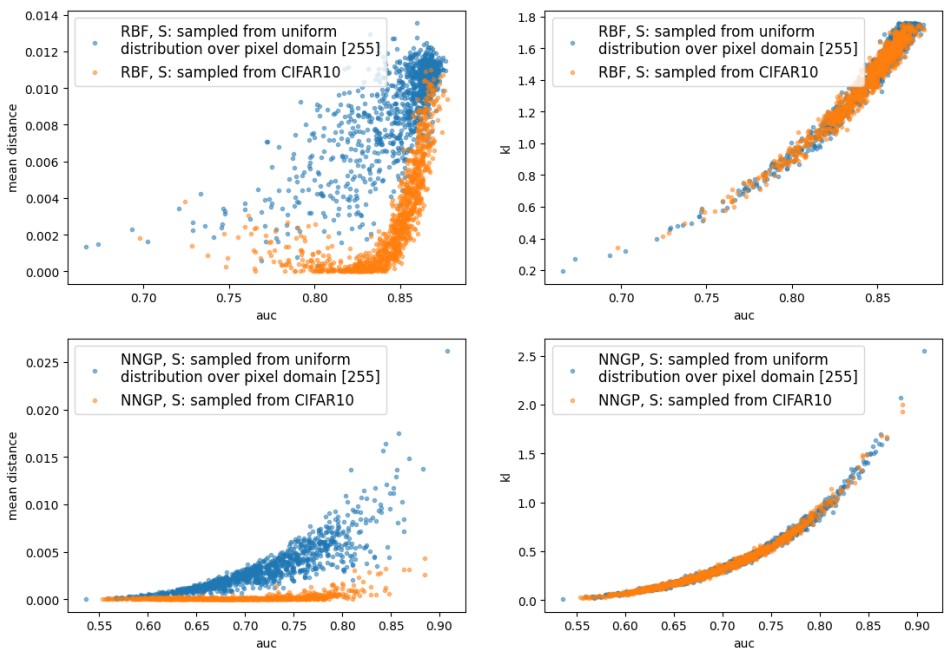

Figure 6: Correlation between the LOOD and MIA success. The leave-one-out dataset $D$ is a class-balanced subset of CIFAR-10 with size 1000, and $D' = D \cup S$ for a randomly chosen differing record $S$. We evaluate over 200 random choices of the differing data record (sampled from the CIFAR-10 dataset or uniform distribution over pixel domain [255].) We also evaluate under different LOOD metrics (mean distance and KL divergence) and different kernel functions (RBF kernel or NNGP kernel for fully connected network with depth 1). We observe that the correlation between KL and AUC (right figures) is consistently stronger and more consistent than the correlation between mean distance LOOD and auc (left figures).

**Example C.2** (Mean distance LOOD is not well-correlated with MIA success). *We show the correlations between Mean distance LOOD and the success of leave-one-out membership inference attack (MIA)* [9] *in Figure 6 (left). We observe that low mean distance and high AUC score occur at the same time, i.e., low Mean distance LOOD does not imply low attack success. This occurs especially when the differing point is sampled from a uniform distribution over the pixel domain* $[0, 255]$.

Another downside of the mean distance LOOD metric is that solely first-order information is incorporated; this results in it being additive in the queries: $M(Q) = \sum_{i=1}^{m} M(Q_i)$. That is, Mean distance LOOD does not allow to incorporate the impact of correlations between multiple queries. Moreover, the mean distance LOOD has a less clear interpretation under certain kernel functions. For example, under the NNGP kernel, a query with large $\ell_2$ norm will also have high Mean Distance LOOD (despite normalized query being the same), due to the homogeneity of NNGP kernel. By contrast, as we show below, KL divergence (relative entropy) LOOD mitigates these limitations.

## C.2 KL Divergence LOOD

To mitigate the downsides of Mean distance LOOD and incorporate higher-order information, we use KL divergence (relative entropy) LOOD.

**Definition C.3** (KL Divergence LOOD). Let $Q \in \mathbb{R}^{d \times q}$ be the set of queried data points, where $d$ is the data feature dimension and $q$ is the number of queries. Let $D, D'$ be a pair of leave-one-out datasets such that $D' = D \cup S$ where $S$ is the differing data point. Let $f_{D,\sigma^2}$ and $f_{D',\sigma^2}$ as coupled Gaussian processes given $D$ and $D'$. Then the Leave-one-out KL distinguishability between $f_{D,\sigma^2}$

---

[9]This is measured by the AUC of the attacker's false positive rate (FPR) versus his true positive rate (TPR).

and $f_{D',\sigma^2}$ on queries $Q$ is as follows,

$$KL(f_{D,\sigma^2}(Q)\|f_{D',\sigma^2}(Q)) = \frac{1}{2}\Big( \log \frac{|\Sigma'(Q)|}{|\Sigma(Q)|} - q + \text{Tr}(\Sigma'(Q)^{-1}\Sigma(Q)) \tag{5}$$

$$+ (\mu(Q) - \mu'(Q))^\top \Sigma'(Q)^{-1}(\mu(Q) - \mu'(Q)) \Big) \tag{6}$$

where for brevity, we denote $\mu(Q) = \mu_{D,\sigma^2}(Q)$, $\mu'(Q) = \mu_{D',\sigma^2}(Q)$, $\Sigma(Q) = \Sigma_{D,\sigma^2}(Q)$, and $\Sigma'(Q) = \Sigma_{D',\sigma^2}(Q)$ as defined in Section 2.

**Example C.4** (KL divergence is better correlated with MIA success). *We show correlations between KL Divergence LOOD and AUC score of the MIA in Figure 6 (right). We observe that KL Divergence LOOD and AUC score are well-correlated with each other.*

***On the assymmetry of KL divergence***. In this paper we restrict our discussions to the case when the base distribution in KL divergence is specified by the predictions on the larger dataset $D'$, which is a reasonable upper bound on the KL divergence with base distribution specified by predictions on the smaller dataset. This is due to the following observations:

1. In numerical experiments (e.g., the middle figure in Figure 7) that $\Sigma(Q)/\Sigma'(Q) \geq 1$ across varying single query $Q$. This is aligned with the intuition that the added information in the larger dataset $D$ reduce its prediction uncertainty (variance).

2. If $\Sigma(Q)/\Sigma'(Q) \geq 1$, then we have that

$$KL(f_{D,\sigma^2}(Q)\|f_{D',\sigma^2}(Q)) \geq KL(f_{D',\sigma^2}(Q)\|f_{D,\sigma^2}(Q))$$

*Proof.* Denote $r = \Sigma(Q)/\Sigma'(Q) \geq 1$, then by Equation (6), for single query $Q$, we have that

$$KL(f_{D,\sigma^2}(Q)\|f_{D',\sigma^2}(Q)) = \frac{1}{2}\left(-\log r - 1 + r + 2M(Q)\Sigma'(Q)^{-1}\right) \tag{7}$$

where $M(Q)$ is the abbreviation for mean distance LOOD as defined in Definition C.1. Similarly, when the base distribution of KL divergence changes, we have that

$$KL(f_{D',\sigma^2}(Q)\|f_{D,\sigma^2}(Q)) = \frac{1}{2}\left(\log r - 1 + \frac{1}{r} + 2M(Q)\Sigma(Q)^{-1}\right) \tag{8}$$

Therefore, we have that

$$KL(f_{D,\sigma^2}(Q)\|f_{D',\sigma^2}(Q)) - KL(f_{D',\sigma^2}(Q)\|f_{D,\sigma^2}(Q))$$

$$= \frac{1}{2}\left(r - \frac{1}{r} - 2\log r + 2M(Q)\Sigma(Q)^{-1}(r-1)\right)$$

For $r \geq 1$, we have that $r - \frac{1}{r} - 2\log r \geq 0$ and $M(Q)\Sigma(Q)^{-1}(r-1) \geq 1$. Therefore, we have that $KL(f_{D,\sigma^2}(Q)\|f_{D',\sigma^2}(Q)) - KL(f_{D',\sigma^2}(Q)\|f_{D,\sigma^2}(Q)) \geq 0$ as long as $r = \Sigma(Q)/\Sigma'(Q) \geq 1$. $\qquad\square$

Therefore, the KL divergence with base prediction distribution specified by the larger dataset is an upper bound for the worst-case KL divergence between prediction distributions on leave-one-out datasets.

## C.3 COMPARISON BETWEEN MEAN DISTANCE LOOD AND KL DIVERGENCE LOOD

***KL Divergence LOOD is better correlated with attack performance***. As observed from Figure 6, KL divergence LOOD exhibits strong correlation to attack performances in terms of AUC score of the corresponding leave-one-out MIA. By contrast, Mean distance LOOD is not well-correlated with MIA AUC score. For example, there exist points that incur small mean distance LOOD while exhibiting high privacy risk in terms of AUC score under leave-one-out MIA. Moreover, from Figure 6 (right), empirically there exists a general threshold of KL Divergence LOOD that implies small AUC (that persists under different choices of kernel function, and dataset distribution). This is reasonable

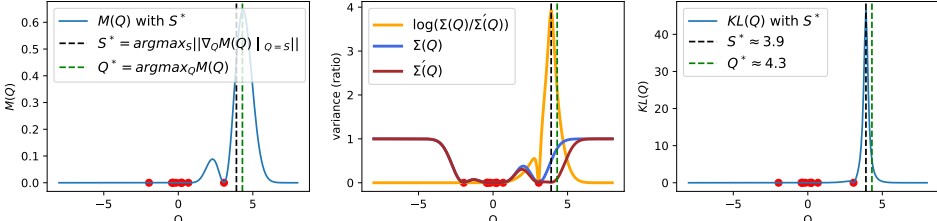

Figure 7: An example on a one-dimensional toy sine training dataset generated as Appendix F.1, where each training data is shown as a red dot. We denote $S^*$ as a crafted differing data record that incurs maximum gradient for the mean distance LOOD objective, which is constructed following the instructions in Algorithm 1. In the left plot, we observe that the optimal query $Q^*$ for mean distance LOOD does not equal the differing point $S^*$. On the contrary, the optimal query for KL LOOD (which incurs maximal information leakage) is the differing point $S^*$ (right plot). This shows the discrepancy between mean distance LOOD and KL LOOD, and shows the inadequacy of using mean distance LOOD to capture information leakage.

since KL divergence is the mean of negative log-likelihood ratio, and is thus closely related to the power of log-likelihood ratio membership inference attack.

***Why the mean is (not) informative for leakage.*** To understand when and why there are discrepancies between the mean distance LOOD and KL LOOD, we numerically investigate the optimal query $Q^*$ that incurs maximal mean distance LOOD. In Figure 7, we observe that although $Q^*$ indeed incurs the largest mean distance among all possible query data, the ratio $\log(\Sigma(Q)/\Sigma'(Q))$ between prediction variance of GP trained on $D$ and that of GP trained on $D'$, is also significantly smaller at query $Q^*$ when compared to the differing point $S$. Consequently, after incorporating the second-order variance information, the KL divergence at query $Q^*$ is significantly lower than the KL divergence at the differing point $S^*$, despite of the fact that $Q^*$ incurs maximal mean distance LOOD. A closer look at Figure 7 middle indicates that the variance ratio peaks at the differing point $S^*$, and drops rapidly as the query becomes farther away from the differing data and the training dataset. Intuitively, this is because $\Sigma'(Q)$, i.e., the variance at the larger dataset, is extremely small at the differing data point $S$. As the query gets further away from the larger training dataset, $\Sigma'(Q)$ then increases rapidly. By contrast, the variance on the smaller dataset $\Sigma(Q)$ is large at the differing data point $S$, but does not change a lot around its neighborhood, thus contributing to low variance ratio. This shows that there exist settings where the second order information contributes significantly to the information leakage, which is not captured by mean distance LOOD (as it only incorporates first-order information).

As another example case of when the ***variance of prediction distribution is non-negligible*** for faithfully capturing information leakage, we investigate under kernel functions that are homogeneous with regard to their input data. Specifically, for a homogeneous kernel function $K$ that satisfifes $K(\lambda x, x') = K(x, \lambda x') = \lambda^\alpha K(x, x')$ for any real number $\lambda \in \mathbb{R}$, we have that

$$\Sigma_{D,\sigma^2}(\lambda Q) = \lambda^{2\alpha}\Sigma_{D,\sigma^2}(Q)$$
$$\mu_{D,\sigma^2}(\lambda Q) = \lambda^\alpha \mu_{D,\sigma^2}(Q),$$

where $\sigma^2 > 0$ denotes the variance of the label noise. Therefore, for any $\lambda \in \mathbb{R}$, we have that

$$M(\lambda Q; D, D') = \lambda^\alpha M(Q; D, D') \tag{9}$$
$$KL(f_{D,\sigma^2}(\lambda Q)\|f_{D',\sigma^2}(\lambda Q)) = KL(f_{D,\sigma^2}(Q)\|f_{D',\sigma^2}(Q)) \tag{10}$$

Consequently, the query that maximizes Mean distance LOOD explodes to infinity for data with unbounded domain. On the contrary, the KL Divergence LOOD is scale-invariant and remains stable as the query scales linearly (because it takes the variance into account, which also grows with the query scale). In this case, the mean distance is not indicative of the actual information leakage as the variance is not a constant and is non-negligible.

In summary, KL LOOD is a metric that is better-suited for the purpose of measuring information leakage, compared to mean distance LOOD. This is because KL LOOD captures the correlation between multiple queries, and incorporates the second-order information of prediction distribution.

# D   DEFERRED PROOFS AND EXPERIMENTS FOR SECTION 4

## D.1   DEFERRED PROOF FOR THEOREM 4.1

In this section, we prove that the differing data record is the stationary point for optimizing the LOOD objective. We will need the following lemma.

**Lemma D.1.** *For $A \in \mathbb{R}^{n \times n}$, $b \in \mathbb{R}^n$ and $c \in \mathbb{R}$, we have that*

$$\begin{pmatrix} A & b \\ b^\top & c \end{pmatrix}^{-1} = \begin{pmatrix} A^{-1} + \alpha^{-1} A^{-1} b b^\top A^{-1} & -\alpha^{-1} A^{-1} b \\ -\alpha^{-1} b^\top A^{-1} & \alpha^{-1} \end{pmatrix} \tag{11}$$

*where $\alpha = c - b^\top A^{-1} b$.*

We now proceed to prove Theorem 4.1

**Theorem 4.1.** Let the kernel function be $K$. Assume that (i) $K(x, x) = 1$ for all training data $x$, i.e., all training data are normalized under kernel function $K$; (ii) $\frac{\partial K(x,x')}{\partial x} \mid_{x'=x} = 0$ for all training data $x$, i.e., the kernel distance between two data records $x$ and $x'$ is stable when they are equal. Then, the differing data is a stationary point for LOOD as a function of query data, as follows

$$\nabla_Q \text{LOOD}(f_{D,\sigma^2}(Q) \| f_{D',\sigma^2}(Q)) \mid_{Q=S} = 0.$$

*Proof.* In this statement, LOOD refers to KL LOOD Definition C.3, therefore by analytical differentiation of Definition C.3, we have that

$$\frac{\partial}{\partial Q} LOOD(f_{D,\sigma^2}(Q) \| f_{D',\sigma^2}(Q)) = \frac{\partial}{\partial Q} KL(f_{D,\sigma^2}(Q) \| f_{D',\sigma^2}(Q)) \tag{12}$$

$$= \left(1 - \frac{\Sigma'(Q)}{\Sigma(Q)}\right) \cdot \frac{\partial}{\partial Q} \left(\frac{\Sigma(Q)}{\Sigma'(Q)}\right)$$

$$+ \frac{\partial \|\mu(Q) - \mu'(Q)\|_2^2}{\partial Q} \Sigma'(Q)^{-1}$$

$$+ \|\mu(Q) - \mu'(Q)\|_2^2 \frac{\partial \Sigma'(Q)^{-1}}{\partial Q}, \tag{13}$$

Note that here $\Sigma(Q)$ and $\Sigma'(Q)$ are both scalars, as we consider single query. We denote the three terms by $F_1(Q)$, $F_2(Q)$, and $F_3(Q)$. We will show that $F_1(S) = 0$ and $F_2(S) = -F_3(S)$ which is sufficient to conclude. Let us start with the first term.

1. $F_1(S) = 0$: Recall from the Gaussian process definition in Section 2 that

$$\Sigma(Q) = K_{QQ} - K_{QD} M_D^{-1} K_{DQ}.$$

where $M_D = K_{DD} + \sigma^2 I$. Therefore,

$$\frac{\partial}{\partial Q} \Sigma(Q) = -2 \frac{\partial K_{QD}}{\partial Q} M_D^{-1} K_{DQ},$$

where $\frac{\partial K_{QD}}{\partial Q} \in \mathbb{R}^{d \times n}$ ($d$ is the dimension of $Q$ and $n$ is the number of datapoints in the training set $D$). A similar formula holds for $\frac{\partial}{\partial Q} \Sigma'(Q)$.

From Lemma D.1, and by using the kernel assumption that $K(S, S) = 1$, we have that

$$M_{D'}^{-1} = \begin{pmatrix} K_{DD} + \sigma^2 I & K_{DS} \\ K_{SD} & 1 + \sigma^2 \end{pmatrix}^{-1} \tag{14}$$

$$= \begin{pmatrix} M_D^{-1} + \alpha^{-1} M_D^{-1} K_{DS} K_{SD} M_D^{-1} & -\alpha^{-1} M_D^{-1} K_{DS} \\ -\alpha^{-1} K_{SD} M_D^{-1} & \alpha^{-1} \end{pmatrix} \tag{15}$$

where $\alpha = 1 + \sigma^2 - K_{SD} M_D^{-1} K_{DS} = \sigma^2 + \Sigma(S)$.

Using the fact that $M_{D'}e_{n+1} = K_{D'S} + \sigma^2 e_{n+1}$ where $e_{n+1} = (0, \ldots, 0, 1)^\top \in \mathbb{R}^{n+1}$, simple calculations yield

$$\begin{aligned}
\Sigma'(S) &= 1 - K_{SD'}M_{D'}^{-1}K_{D'S} \\
&= \sigma^2(1 - \sigma^2 e_{n+1}^\top M_{D'}^{-1}e_{n+1}) \\
&= \sigma^2(1 - \sigma^2 \alpha^{-1}).
\end{aligned}$$

We now have all the ingredients for the first term. To alleviate the notation, we denote by $\mathring{K}_{SD} = \frac{\partial}{\partial Q} K_{QD} \mid_{Q=S}$.

We obtain

$$\frac{\partial}{\partial Q} \left( \frac{\Sigma(Q)}{\Sigma'(Q)} \right) \mid_{Q=S} = \frac{-2\mathring{K}_{SD}M_D^{-1}K_{DS}}{\Sigma'(S)} + \frac{2\Sigma(S)\mathring{K}_{SD'}M_{D'}^{-1}K_{D'S}}{\Sigma'(S)^2}.$$

A key observation here is the fact that the last entry of $\mathring{K}_{SD'}$ is 0 by assumption. Using the formula above for $M_{D'}^{-1}$, we obtain

$$\begin{aligned}
\mathring{K}_{SD'}M_{D'}^{-1}K_{D'S} &= -\sigma^2 \mathring{K}_{SD'}M_{D'}^{-1}e_{n+1} \\
&= -\sigma^2 \mathring{K}_{SD}(-\alpha^{-1}M_D^{-1}K_{DS}) \\
&= \alpha^{-1}\sigma^2 \mathring{K}_{SD}M_D^{-1}K_{DS}).
\end{aligned}$$

Now observe that $\alpha^{-1}\sigma^2\Sigma(S)\Sigma'(S)^{-1} = \alpha^{-1}\sigma^2(\alpha - \sigma^2)\sigma^{-2}(1 - \sigma^2\alpha^{-1})^{-1} = 1$, which concludes the proof for the first term.

2. $F_2(S) + F_3(S) = 0$:

Let us start by simplifying $F_2(S) = \frac{\partial \|\mu(Q) - \mu'(Q)\|_2^2}{\partial Q} \mid_{Q=S} \Sigma'(S)^{-1}$. The derivative here is that of $M(Q)$, and we have

$$\frac{\partial \|\mu(Q) - \mu'(Q)\|_2^2}{\partial Q} \mid_{Q=S} = 2(\mu(S) - \mu'(S))(\mathring{K}_{SD}M_D^{-1}y_D - \mathring{K}_{SD'}M_{D'}^{-1}y_{D'}).$$

Using the formula of $M_{D'}^{-1}$ and observe that the last entry of $\mathring{K}_{SD}$ is zero, we obtain

$$\frac{\partial \|\mu(Q) - \mu'(Q)\|_2^2}{\partial Q} \mid_{Q=S} = 2\alpha^{-1}(\mu(S) - \mu'(S)) \left(y_S - K_{SD}M_D^{-1}y_D\right) \mathring{K}_{SD}M_D^{-1}K_{DS}.$$

Moreover, we have that

$$\begin{aligned}
\mu(S) - \mu'(S) &= K_{SD}M_D^{-1}y_D - K_{SD'}M_{D'}^{-1}y_{D'} & (16) \\
&= K_{SD}M_D^{-1}y_D - e_{n+1}^\top(M_{D'} - \sigma^2 I)M_{D'}^{-1}y_{D'} & (17) \\
&= K_{SD}M_D^{-1}y_D - y_S + \sigma^2 e_{n+1}^\top M_{D'}^{-1}y_{D'} & (18) \\
&= K_{SD}M_D^{-1}y_D - y_S + \sigma^2(-\alpha^{-1}K_{SD}M_D^{-1}y_D + \alpha^{-1}y_S) & (19) \\
&= (\sigma^2\alpha^{-1} - 1)(y_S - K_{SD}M_D^{-1}y_D). & (20)
\end{aligned}$$

Therefore

$$F_2(S) = 2\alpha^{-1}(\sigma^2\alpha^{-1} - 1)\Sigma'(S)^{-1} \left(y_S - K_{SD}M_D^{-1}y_D\right)^2 \mathring{K}_{SD}M_D^{-1}K_{DS} \quad (21)$$

Let us now deal with $F_3(S) = (\mu(S) - \mu'(S))^2 \frac{\partial \Sigma'(Q)^{-1}}{\partial Q} \mid_{Q=S}$. We have that

$$\frac{\partial \Sigma'(Q)^{-1}}{\partial Q} \mid_{Q=S} = 2\Sigma'(S)^{-2} \mathring{K}_{SD'} M_{D'}^{-1} K_{D'S}$$

$$= -2\sigma^2 \Sigma'(S)^{-2} \mathring{K}_{SD'} M_{D'}^{-1} e_{n+1}$$

$$= -2\sigma^2 \Sigma'(S)^{-2} \mathring{K}_{SD}(-\alpha^{-1} M_D^{-1} K_{DS})$$

$$= 2\alpha^{-1}\sigma^2 \Sigma'(S)^{-2} \mathring{K}_{SD} M_D^{-1} K_{DS}. \tag{22}$$

By plugging Equation (22) and 20 into $F_3(S) = (\mu(S) - \mu'(S))^2 \frac{\partial \Sigma'(Q)^{-1}}{\partial Q} \mid_{Q=S}$, we prove that

$$F_3(S) = 2\alpha^{-1}\sigma^2 (\sigma^2 \alpha^{-1} - 1)^2 \Sigma'(S)^{-2} \left(y_S - K_{SD} M_D^{-1} y_D\right)^2 \mathring{K}_{SD'} M_D^{-1} K_{DS}.$$

We conclude by observing that $\sigma^2(\sigma^2 \alpha^{-1} - 1)\Sigma'(S)^{-1} = \sigma^2(\sigma^2 \alpha^{-1} - 1)\sigma^{-2}(1 - \sigma^2 \alpha^{-1})^{-1} = -1$.

$\square$

## D.2 Proofs for regularity conditions of commonly used kernel functions

We first show that both the isotropic kernel and the correlation kernel satisfies the condition of Theorem 4.1. We then show that the RBF kernel, resp. the NNGP kernel on a sphere, is an isotropic kernel, resp. a correlation kernel.

**Proposition D.2** (Isotropic kernels satisfy conditions in Theorem 4.1). *Assume that the kernel function $K$ is isotropic, i.e. there exists a continuously differentiable function $g$ such that $K(x, y) = g(\|x - y\|)$ for all $x, y$. Assume that $g$ satisfies $z^{-1} g'(z)$ is bounded[10] and $g(0) = 1$. Then, $K$ satisfies the conditions of Theorem 4.1.*

*Proof.* It is straightforward that $K(x, x) = 1$ for all $x$. Simple calculations yield

$$\frac{\partial K(x, y)}{\partial x} = \frac{g'(\|x - y\|)}{\|x - y\|}(x - y).$$

By assumption, the term $\frac{g'(\|x-y\|)}{\|x-y\|}$ is bounded for all $x \neq y$ and also in the limit $x \to y$. Therefore, it is easy to see that

$$\frac{\partial K(x, y)}{\partial x} \mid_{x=y} = 0.$$

More rigorously, the partial derivative above exists by continuation of the function $z \to \frac{g'(z)}{z} z$ near $0^+$. $\square$

**Proposition D.3** (Correlation kernels satisfy conditions in Theorem 4.1). *Assume that the kernel function $K$ is a correlation kernel, i.e. there exists a function $g$ such that $K(x, y) = g\left(\frac{\langle x, y \rangle}{\|x\| \|y\|}\right)$ for all $x, y \neq 0$. Assume that $g(1) = 1$. Then, $K$ satisfies the conditions of Theorem 4.1.*

*Proof.* The first condition is satisfied by assumption on $g$. For the second condition, observe that

$$\frac{\partial K(x, y)}{\partial x} = \frac{1}{\|x\| \|y\|} \left(y - \frac{\langle x, y \rangle}{\|x\|^2} x\right) g'\left(\frac{\langle x, y \rangle}{\|x\| \|y\|}\right),$$

which yields $\frac{\partial K(x, y)}{\partial x} \mid_{x=y} = 0$. $\square$

We have the following result for the NNGP kernel.

---

[10]Actually, only the boundedness near zero is needed.

**Proposition D.4** (NNGP kernel on the sphere satisfies conditions in Theorem Theorem 4.1). *The NNGP kernel function is similar to the correlation kernel. For ReLU activation function, it has the form*

$$K(x, y) = \|x\|\|y\|g\left(\frac{\langle x, y\rangle}{\|x\|\|y\|}\right).$$

*This kernel function does not satisfy the conditions of Theorem 4.1 and therefore we cannot conclude that the differing point is a stationary point. Besides, in practice, NNGP is used with normalized data in order to avoid any numerical issues; the datapoints are normalized to have $\|x\| = r$ for some fixed $r > 0$. With this formulation, we can actually show that the kernel $K$ restricted to the sphere satisfies the conditions of Theorem 4.1. However, as one might guess, it is not straightforward to compute a "derivative on the sphere" as this requires a generalized definition of the derivative on manifolds (the coordinates are not free). To avoid dealing with unnecessary complications, we can bypass this problem by considering the spherical coordinates instead. For $x \in \mathbb{R}^d$, there exists a unique set of parameters $(r, \varphi_1, \varphi_2, \ldots, \varphi_{d-1}) \in \mathbb{R}^d$ s.t.*

$$\begin{cases} x_1 = r\cos(\varphi_1) \\ x_2 = r\sin(\varphi_1)\cos(\varphi_2) \\ x_3 = r\sin(\varphi_1)\sin(\varphi_2)\cos(\varphi_3) \\ \vdots \\ x_{d-1} = r\sin(\varphi_1)\sin(\varphi_2)\ldots\sin(\varphi_{d-2})\cos(\varphi_{d-1}) \\ x_d = r\sin(\varphi_1)\sin(\varphi_2)\ldots\sin(\varphi_{d-2})\sin(\varphi_{d-1}). \end{cases}$$

*This is a generalization of the polar coordinates in $2D$.*

*Without loss of generality, let us assume that $r = 1$ and let $\boldsymbol{\varphi} = (\varphi_1, \varphi_2, \ldots, \varphi_{d-1})$. In this case, the NNGP kernel, evaluated at $(\boldsymbol{\varphi}, \boldsymbol{\varphi}')$ is given by*

$$K(\boldsymbol{\varphi}, \boldsymbol{\varphi}') = g\left(\sum_{i=1}^{d-1}\left[\prod_{j=1}^{i-1}\sin(\varphi_j)\sin(\varphi_j')\right]\cos(\varphi_i)\cos(\varphi_i') + \prod_{j=1}^{d-1}\sin(\varphi_j)\sin(\varphi_j')\right),$$

*with the convention $\prod_{i=1}^{0}\cdot = 1$.*

*Consider the spherical coordinates introduced above, and assume that $r = 1$. Then, the NNGP kernel given by $K(\boldsymbol{\varphi}, \boldsymbol{\varphi}')$ satisfies the conditions of Theorem 4.1.*

*Proof.* Computing the derivative with respect to $\boldsymbol{\varphi}$ is equivalent to computing the derivative on the unit sphere. For $l \in \{1, \ldots, d-1\}$, we have

$$\frac{\partial K(\boldsymbol{\varphi}, \boldsymbol{\varphi}')}{\partial \varphi_l} = J(\boldsymbol{\varphi}, \boldsymbol{\varphi}') \times g'\left(\sum_{i=1}^{d-1}\left[\prod_{j=1}^{i-1}\sin(\varphi_j)\sin(\varphi_j')\right]\cos(\varphi_i)\cos(\varphi_i') + \prod_{j=1}^{d-1}\sin(\varphi_j)\sin(\varphi_j')\right),$$

where

$$J(\boldsymbol{\varphi}, \boldsymbol{\varphi}') = \sum_{i=l+1}^{d-1}\left[\cos(\varphi_l)\prod_{\substack{j=1\\j\neq l}}^{i-1}\sin(\varphi_j)\right]\left[\prod_{j=1}^{i-1}\sin(\varphi_j')\right]\cos(\varphi_i)\cos(\varphi_i')$$

$$+ \cos(\varphi_l)\left[\prod_{\substack{j=1\\j\neq l}}^{d-1}\sin(\varphi_j)\right]\left[\prod_{j=1}^{d-1}\sin(\varphi_j')\right] - \left[\prod_{j=1}^{l-1}\sin(\varphi_j)\right]\left[\prod_{j=1}^{l-1}\sin(\varphi_j')\right]\sin(\varphi_l)\cos(\varphi_l').$$

When $\varphi = \varphi'$, we obtain

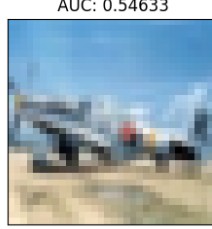 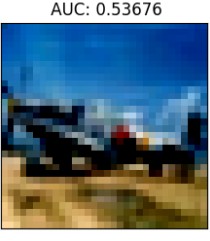 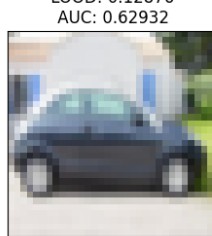 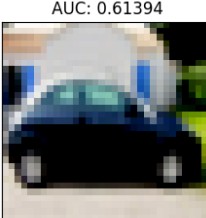

(a) A run of LOOD optimization under NNGP kernel for fully connected network with depth 5

(b) A run of LOOD optimization under NNGP kernel for CNN with depth 5

Figure 8: Additional LOOD optimization results under NNGP kernels with different architectures. We evaluate on class-balanced subset of CIFAR-10 dataset of size 1000. The optimized query consistently has similar LOOD and MIA AUC score to the differing data record.

$$J(\boldsymbol{\varphi}, \boldsymbol{\varphi}) = \frac{\cos(\varphi_l)}{\sin(\varphi_l)} \sum_{i=l+1}^{d-1} \left[ \prod_{j=1}^{i-1} \sin(\varphi_j)^2 \right] \cos(\varphi_i)^2 + \frac{\cos(\varphi_l)}{\sin(\varphi_l)} \left[ \prod_{j=1}^{d-1} \sin(\varphi_j)^2 \right]$$

$$- \frac{\cos(\varphi_l)}{\sin(\varphi_l)} \left[ \prod_{j=1}^{l} \sin(\varphi_j)^2 \right].$$

By observing that

$$\sum_{i=l+1}^{d-1} \left[ \prod_{j=1}^{i-1} \sin(\varphi_j)^2 \right] \cos(\varphi_i)^2 + \left[ \prod_{\substack{j=1 \\ j \neq l}}^{d-1} \sin(\varphi_j)^2 \right] = \prod_{j=1}^{l} \sin(\varphi_j)^2,$$

we conclude that $\frac{\partial K(\boldsymbol{\varphi}, \boldsymbol{\varphi}')}{\partial \varphi_l} \big|_{\boldsymbol{\varphi} = \boldsymbol{\varphi}'} = 0$ for all $l$.

$\square$

### D.3 MORE EXAMPLES FOR QUERY OPTIMIZATION UNDER NNGP KERNEL FUNCTIONS

We show the optimized query for NNGP kernel under different model architectures, including fully connected network and convolutional neural network with various depth, in Figure 8.

To further understand the query optimization process in details, we visualize an example LOOD optimization process below in Figure 9.

### D.4 WHEN THE DIFFERING POINT IS NOT THE OPTIMAL QUERY

In practice data-dependent normalization is used prior to training, such as batch and layer normalization. Note that this normalization of the pair of leave-one-out datasets results in them differing in more than one record [11] Then the optimized query data record obtains a higher LOOD than the LOOD of the differing points; we show this in Figure 10; thus in such a setting there is value for the attacker to optimise the query; this is consistent with results from (Wen et al., 2022) that exploit the training data augmentation.

---

[11]During inference, the same test record will again go through a normalization step that depends on the training dataset before being passed as a query the corresponding Gaussian process.

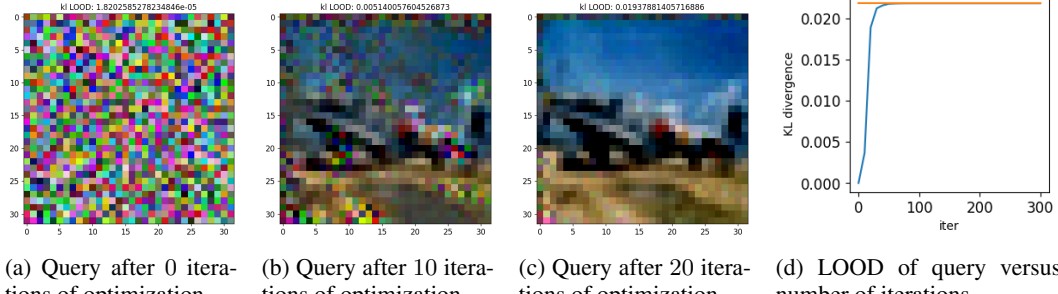

(a) Query after 0 iterations of optimization

(b) Query after 10 iterations of optimization

(c) Query after 20 iterations of optimization

(d) LOOD of query versus number of iterations

Figure 9: Illustration of an example LOOD optimization process at different number of optimization iterations. We evaluate on a class-balanced subset of CIFAR-10 dataset of size 1000 and NNGP kernel for fully connected network with depth 5. The query is initialized by uniform sampling from the pixel domain, and starts recovering the differing data record as the optimization proceeds. We plot the optimized queries after 0, 10 and 20 iterations and show the convergence of LOOD for query with regard to the increasing number of iterations for LOOD optimization.

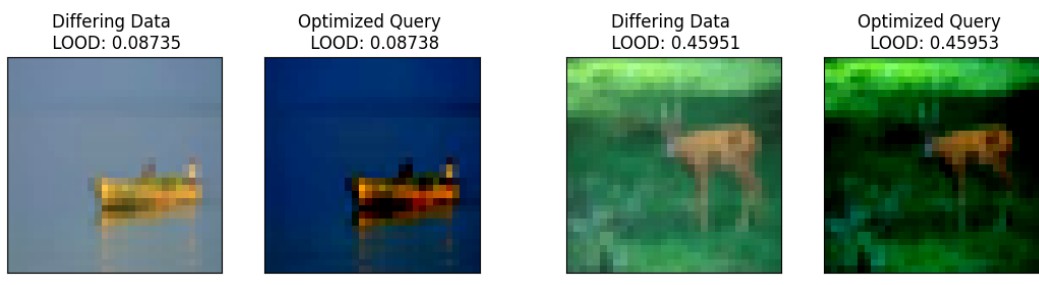

(a) Differing data and optimized query in run one

(b) Differing data and optimized query in run two

Figure 10: Example runs of LOOD optimization under data-dependent training data normalization (that uses per-channel mean and standard deviation of training dataset). The optimized query has a slightly higher LOOD than differing point, and visually has darker color. Our setup uses 500 training data points of the CIFAR10 dataset (50 per class) and NNGP specified by CNN with depth 5.

# E   ANALYZING LOOD FOR MULTIPLE QUERIES AND LEAVE-ONE-GROUP-OUT SETTING

## E.1   LOOD UNDER MULTIPLE QUERIES

We now turn to a more general question of quantifying the LOOD when having a model answer **multiple queries** $\{Q_1, ..., Q_q\}$ rather than just one query $Q$ (this corresponds to the practical setting of e.g. having API access to a model that one can query multiple times). In this setting, a natural question is: as the number of queries grows, does the distinguishability between leave-one-out models' predictions get stronger (when compared to the single query setting)? Our empirical observations indicate that the significant information regarding the differing data point resides within its prediction on that data record. Notably, the optimal single query $Q = S$ exhibits a higher or equal degree of leakage compared to any other query set.

Specifically, to show that the optimal single query $Q = S$ exhibits a higher or equal degree of leakage compared to any other query set, we performed multi-query optimization $\max_{Q_1, \cdots, Q_q} LOOD(Q_1, \cdots, Q_q; D, D')$, which uses gradient descent to optimize the LOOD across $q$ queries, for varying numbers of queries $q = 1, 2, 4, 8$. We evaluated this setting under the following conditions: NNGP using a fully connected network with depths of $2, 4, 6, 8, 10$; using 500 random selections of leave-one-out datasets (where the larger dataset is of size 10000) sampled from the 'car' and 'plane' classes of the CIFAR10 dataset. Our observations indicate that, in all

evaluation trials, the LOOD values obtained from the optimized $m$ queries are lower than the LOOD value from the single differing data record.

## E.2 LEAVE-ONE-GROUP OUT DISTINGUISHABILITY

Here we extend our analysis to the **leave-one group-out** setting, which involves two datasets, $D$ and $D'$, that differ in a *group* of $s$ data points $\{S_1, ..., S_s\}$. There are no specific restrictions on how these points are selected. In this setup, we evaluate the model using queries $\{Q_1, ..., Q_q\}$, where $q \geq 1$. The leave-one group-out approach can provide insights into privacy concerns when adding a group to the training set, such as in datasets containing records from members of the same organization or family Dwork et al. (2014). Additionally, this setting can shed light on the success of data poisoning attacks, where an adversary adds a set of points to the training set to manipulate the model or induce wrong predictions. Here, we are interested in constructing and studying queries that result in the highest information leakage about the group, i.e., where the model's predictions are most affected under the leave-one group-out setting.

Similar to our previous approach, we run maximization of LOOD to identify the queries that optimally distinguishes the leave-one group-out models. Figure 3 presents the results for single-query ($q = 1$) setting and multi-query ($q = 3$) setting, with a differing group size $|S| = 3$. We have conducted in total 10 rounds of experiments, where in each round we randomly select a fixed group of differing datapoints and then perform 10 independent optimization runs (starting from different initializations). Among all the runs, $80\%$ of them converged, in which we observe that the optimal queries are visually similar to the members of the differing group, but the chance of recovering them varies based on how much each individual data point is exposed through a single query (by measuring LOOD when querying the point). We observe that among the points in set $S$, the differing data point with the highest single-query LOOD was recovered $61\%$ of the times, whereas the one with the lowest single-query LOOD was recovered only $8\%$ of the times. This can be leveraged for data reconstruction attacks where the adversary has access to part of the train dataset of the model.

## F OPTIMAL QUERY UNDER MEAN DISTANCE AND DEFERRED PROOFS FOR SECTION 5

We discuss here the computation of the optimal query under mean distance. This is equivalent to the question of where the function is most influenced by a change in the dataset.

### F.1 DIFFERING POINT IS IN GENERAL NOT OPTIMAL FOR MEAN DISTANCE LOOD

***Numerically identifying cases where the differing point is not optimal.*** We define an algorithm that finds non-optimal differing points $S$ such that when optimizing the query $Q$, the resulting query is different from $S$. When the differing point $S$ is not the optimal query, the gradient of $M(Q)$ evaluated at $S$ is non-zero. Hence, intuitively, finding points $S$ such that $\|\nabla_Q M(Q) \mid_{Q=S}\|$ is maximal should yield points $S$ for which the optimal query is not the differing point (as there is room for optimising the query due to its positive gradient). We describe this algorithm in Algorithm 1.

**Algorithm 1** Identifying non-optimal differing points $S$

---

Given $D$ and $\sigma$, we solve the optimization problem $S^* = \text{argmax}_S \|\nabla_Q M(Q) \mid_{Q=S}\|$ to identify a non-optimal differing point. (We use gradient descent to solve this problem.)

---

***A toy example of where the differing point is not optimal.*** To assess the efficacy of Algorithm 1, we consider a simple example in the one-dimensional case. The setup is as follows: **(1)** Data $D = \{(x_i, y_i), i = 1, \dots, 10\}$ is generated using the following distribution: $x_i \sim \mathcal{N}(0, 1)$, $y_i = \sin(x_i)$. **(2)** Noise parameter is given by $\sigma^2 = 0.01$. **(3)** Kernel is given by $K(x, x') = \exp(-(x - x')^2/2)$ (RBF kernel). **(4)** In Algorithm 1, only $S_x$ is optimized, the label $S_y$ is fixed to $S_y = \sin(S_x)$.

We show in Figure 11a the gradient of $M(Q)$ evaluated at $S$ for varying $S$, and highlight the point $S^*$ (the result of Algorithm 1) that has the largest non-stationarity for mean distance LOOD objective. Additionally, if we look at the query with maximal LOOD under differing data $S = S^*$, the optimal query $Q^*$ that we found is indeed different from the differing point $S^*$, as shown in Figure 11b.

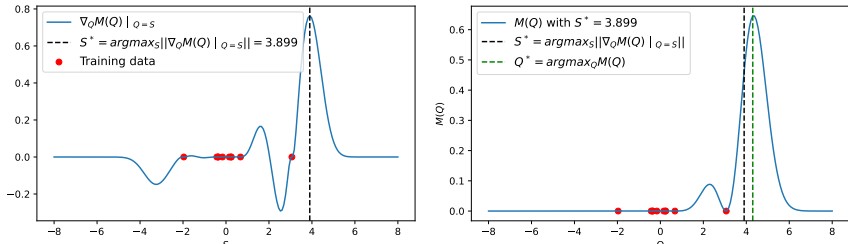

(a) The gradient of $M(Q)$ evaluated at $S$.     (b) The mean distance $M(Q)$ under $S^*$.

Figure 11: We plot the stationary condition indicator function $\nabla_Q M(Q) \mid_{Q=S}$ of LOOD when query $Q$ equals the differing data $S$, over different values of differing data $S$ with $S_x \in [-5, 5]$. This allows us to run Algorithm 1 to identify a differing point $S^* = \max_S \|\nabla_Q M(Q) \mid_{Q=S} \|$ that has the highest objective (maximal non-stationarity), as shown in Algorithm 1 (where $S^* = 3.899$). In Figure 11b, we validate that under the optimized differing data $S^*$, the query $Q^*$ that maximizes LOOD (over leave-one-out datasets that differ in $S^*$) is indeed not the differing point $S^*$.

Let us now look at what happens when we choose other differing points $S \neq S^*$. Figure 12 shows

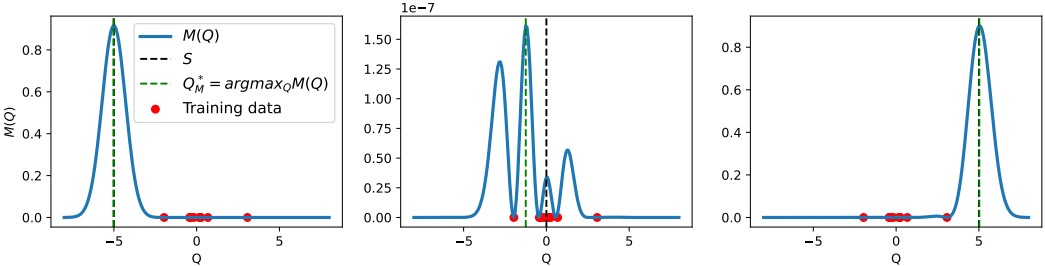

Figure 12: The plot of $M(Q)$ for varying $Q$ and $S$. We also show the optimal query $Q^*$ and the differing point $S$.

the optimal query for three different choices of $S$. When $S$ is *far* from the training data $D$ ($S_x \in \{-5, 5\}$), the optimal query approximately coincides with the differing point $S$. When $S$ is close to the training data ($S_x = 0$), the behaviour of the optimal query is non-trivial, but it should be noted that the value of the MSE is very small in this case.

***Explaining when the differing point is suboptimal for Mean Distance LOOD***. We now give a theoretical explanation regarding why the differing point is suboptimal for mean distance LOOD when it is close (but not so close) to the training dataset (as observed in Figure 12 middle plot). We first give a proposition that shows that the gradient of $M(Q)$ is large when the differing point $S$ is close (but not so close) to the training dataset $D$.

***Proposition 5.1.*** Consider a dataset $D$, a differing point $S$, and a kernel function $K$ satisfying the same conditions of Theorem 4.1. Then, we have that

$$\nabla_Q M(Q) \mid_{Q=S} = -\alpha^{-2}(1 - K_{SD}M_D^{-1}K_{DS})(y_S - K_{SD}M_D^{-1}y_D)^2 \mathring{K}_{SD}M_D^{-1}K_{DS}, \quad (23)$$

where $M(Q)$ denotes the mean distance LOOD as defined in Definition C.1, $K_{SD} \in \mathbb{R}^{1 \times |D|}$ is the kernel matrix between data point $S$ and dataset $D$, $K_{DS} = K_{SD}^\top$, $\mathring{K}_{SD} = \frac{\partial}{\partial Q}K_{QD} \mid_{Q=S}$, $M_D = K_{DD} + \sigma^2 I$, and $\alpha = 1 - K_{SD}M_D^{-1}K_{DS} + \sigma^2$.

*Proof.* The left-hand-side of Equation (23) is $\nabla_Q M(Q) = \frac{\partial}{\partial Q}\|\mu(Q) - \mu'(Q)\|_2^2$ by definition of the mean distance LOOD $M(Q)$ in Definition C.1. This term occurs in the term $F_2(Q)$ in Equation (13), in the proof for Theorem 4.1. Specifically, we have $\nabla_Q M(Q) = F_2(S) \cdot \Sigma'(S)$. The detailed expression for $F_2(S)$ is computed in line Equation (21) during the proof for Theorem 4.1. Therefore, by multiplying $F_2(S)$ in Equation (21) with $\Sigma'(S)$, we obtain the expression for $\nabla_Q M(Q)$ in the statement Equation (23). □

As a result, it should be generally expected that $\nabla_Q M(Q)\mid_{Q=S} \neq 0$. We can further see that when $S$ is far from the data in the sense that $K_{SD} \approx 0$, the gradient is also close to 0. In the opposite case where $S$ is close to $D$, e.g. satisfying $y_S \approx K_{SD} M_D^{-1} y_D$, then the gradient is also close to zero. Between the two cases, there exists a region where the gradient norm is maximal.

### F.2 DIFFERING POINT IS OPTIMAL IN MEAN DISTANCE LOOD WHEN IT IS FAR FROM THE DATASET

***Explaining when the differing point is optimal for Mean Distance LOOD***. We now prove that the differing point is (nearly) optimal when it is far from remaining dataset. We assume the inputs belong to some compact set $C \in \mathbb{R}^d$.

**Definition F.1** (Isotropic kernels). We say that a kernel function $K : C \times C \to \mathbb{R}$ is isotropic is there exists a function $g : \mathbb{R} \to \mathbb{R}$ such that $K(x, x') = g(\|x - x'\|)$ for all $x, x' \in C$.

Examples includes RBF kernel, Laplace kernel, etc. For this type of kernel, we have the following result.

**Lemma F.2.** *Let $K$ be an isotropic kernel, and assume that its corresponding function $g$ is differentiable on $C$ (or just the interior of $C$). Then, for all $Q \in C$, we have that*

$$\|\nabla_Q M(Q)\| \leq \zeta \left(d'(Q, S) + d'(Q, D)\right) \max(d(Q, S), d(Q, D)),$$

*where*

- $D = \{D_1, \ldots, D_n\}$, $D' = D \cup \{S\}$
- $d'(Q, D) = \max_{i \in [n]} |g'(\|Q - D_i\|)|$ *and* $d'(Q, S) = g'(\|Q - S\|)$
- $d(Q, D) = \max_{i \in [n]} |g(\|Q - D_i\|)|$ *and* $d(Q, S) = g(\|Q - S\|)$
- $\zeta > 0$ *is a constant that depends only on $D$.*

*Proof.* Let $Q \in C$. For the sake of simplification, we use the following notation: $u = (K_{DD} + \sigma^2 I)^{-1} y_D$, and $u' = (K_{D'D'} + \sigma^2 I)^{-1} y_{D'}$.

Simple calculations yield

$$\nabla_Q M(Q) = (K_{QD'} u' - K_{QD} u) \left( \sum_{i=1}^{n} (u_i' - u_i) \nabla_Q K_{QD_i} + u_{n+1}' \nabla_Q K_{Q,S} \right).$$

The first term is upperbounded by $\|u_{[n]}' - u\| d(Q, D) + |u_{n+1}'| d(Q, S)$, where $u_{[n]}'$ is the vector containing the first $n$ coordinates of $u'$, and $u_i$ refers to the $i^{th}$ coordinate of the vector $u$.

Now observe that $\|u'\|$ is bounded for all $S$, provided the data labels $y$ are bounded. For this, it suffices to see that $\|(K_{D'D'} + \sigma^2 I)^{-1}\|_2 \leq \sigma^{-2}$ because of the positive semi-definiteness of $K_{D'D'}$.

Moreover, we have $\nabla_Q K_{QD_i} = g'(\|Q - D_i\|) \frac{Q - D_i}{\|Q - D_i\|}$, which implies that $\|\nabla_Q K_{QD_i}\| = |g'(\|Q - D_i\|)|$. Therefore, using the triangular inequality, we obtain

$$\left\| \sum_{i=1}^{n} (u_i' - u_i) \nabla_Q K_{QD_i} + u_{n+1}' \nabla_Q K_{Q,S} \right\| \leq \|u_{[n]}' - u\|_1 d'(Q, D) + |u_{n+1}'| |g'(\|Q - S\|)|.$$

The existence of $\zeta$ is trivial. $\square$

From Lemma F.2, we have the following corollary.

**Corollary F.3.** *Assume that $\sup_{z \in \mathbb{R}} g(z) < \infty$, $\lim_{|z| \to \infty} g'(z) = 0$ and $g'(0) = 0$. Then, provided that $C$ is large enough, for any $D = \{D_1, \ldots, D_n\}$ and $\epsilon > 0$, there exists a point $S$, such that if we define $D' = D \cup \{S\}$, then $\|\nabla_Q M(S)\| \leq \epsilon$.*
*That is, if we choose $S$ far enough from the data $D$, then the point $S$ is close to a stationary point.*

Although this shows that gradient is small at $S$ (for $S$ far enough from the data $D$), it does not say whether this is close to a local minimum or maximum. T determine the nature of the stationary points in this region, an analysis of the second-order geometry is needed. More precisely, we would like to understand the behaviour of the Hessian matrix in a neighbourhood of $S$. In the next result, we show that the Hessian matrix is strictly negative around $S$ (for $S$ sufficiently far from the data $D$), which confirms that any stationary point in the neighbourhood of $S$ should be a local maximum.

**Lemma F.4** (Second-order analysis). *Assume that $g(0) > 0$, $\lim_{z \to 0} z^{-1}g'(z) = l_-$, where $l_-$ is a negative constant that depends on the kernel function, and that $\lim_{z \to \infty} z^{-1}g'(z) = 0$. Then, provided that $C$ is large enough, for any $D = \{D_1, \ldots, D_n\}$ and $\epsilon > 0$, there exists a point $S$, such that if we define $D' = D \cup \{S\}$, then there exists $\eta > 0$ satisfying: for all $Q \in \mathcal{B}(S, \eta), \nabla_Q^2 M(Q) \prec 0$.*

***Lemma F.4.*** Assume that $g(0) > 0$, $\lim_{z \to 0} z^{-1}g'(z) = l_-$, where $l_- < 0$ is a negative constant that depends on the kernel function, and that $\lim_{z \to \infty} z^{-1}g'(z) = 0$. Then, provided that $C$ is large enough, for any $D = \{D_1, \ldots, D_n\}$ and $\epsilon > 0$, there exists a point $S$, such that if we define $D' = D \cup \{S\}$, then there exists $\eta > 0$ satisfying: for all $Q \in \mathcal{B}(S, \eta), \nabla_Q^2 M(Q) \prec 0$ is a negatively definite matrix.

*Proof.* Let $Q \in C$. Let $v(Q) = K_{QD'}u' - K_{QT}u$. The Hessian is given by

$$\nabla_Q^2 M(Q) = v(Q)\nabla_Q^2 v(Q) + \nabla_Q v(Q)\nabla_Q v(Q)^\top$$

where $\nabla_Q^2 v(Q) = \sum_{i=1}^n (u_i' - u_i)\nabla_Q^2 K_{QD_i} + u_{n+1}' \nabla_Q^2 K_{Q,S}$ and $\nabla_Q v(Q) = \sum_{i=1}^n (u_i' - u_i)\nabla_Q K_{QD_i} + u_{n+1}' \nabla_Q K_{Q,S}$. By Lemma F.2, we have that for $S$ far enough from the dataset $D$, we have that $\|\nabla_Q v(S)\| \leq \epsilon$. Therefore, the Hessian at $S$ simplilfies to the following.

$$\nabla_Q^2|_{Q=S}M(Q) \leq v(Q)\left(\sum_{i=1}^n (u_i' - u_i)\nabla_Q^2 K_{QD_i} + u_{n+1}' \nabla_Q^2 K_{Q,S}\right) + \epsilon^2.$$

For $i \in [n]$, simple calculations yield

$$\nabla_Q^2 K_{QD_i} = \alpha(Q, D_i)\frac{(Q - D_i)(Q - D_i)^\top}{\|Q - D_i\|^2} + \frac{g'(\|Q - D_i\|)}{\|Q - D_i\|}I,$$

where $\alpha(Q, D_i) = g''(\|Q - D_i\|) - \frac{g'(\|Q-D_i\|)}{\|Q-D_i\|}$.

Notice that $u_{n+1}' \approx y_S - K_{SD}u$. Without loss of generality, assume that $y_S > 0$. For $S$ far enough from the dataset $D$, we then have $u_{n+1}' > 0$ and $v(S) \approx y_S > 0$. Let us now see what happens to the Hessian. For such $S$, and for any $\epsilon > 0$, there exists a neighbourhood $\mathcal{B}(S, \eta), \eta > 0$, such that for all $Q \in \mathcal{B}(S, \eta)$, we have $\frac{|g'(\|Q-D_i\|)|}{\|Q-D_i\|} < \epsilon$, and $|\alpha(Q, D_i)| < \epsilon$. This ensures that $\|\nabla_Q^2 K_{QD_i}\|_F \leq 2\epsilon$ where $\|.\|_F$ denotes the Frobenius norm of a matrix. On the other hand, we have $\nabla_Q^2 K_{Q,S} = \lim_{z \to 0} \frac{g'(z)}{z}I = l_- I$. It remains to deal with $v(Q)$. Observe that for $S$ far enough from the data $D$, and $Q$ close enough to $S$, $v(Q)$ is close to $v(S) \approx y_S > 0$. Thus, taking $\epsilon$ small enough, there exists $\eta$ satisfying the requirements, which concludes the proof. $\square$

Lemma F.4 shows that whenever the point $S$ is isolated from the dataset $D$, the Hessian is negative in a neighbourhood of $S$, suggesting that any local stationary point can only be a local maximum. This is true for isotropic kernels such as the Radial Basis Function kernel (RBF) and the Laplace kernel. As future work, Lemma F.2 and Lemma F.4 have important computational implications for the search for the 'worst' poisonous point $S$. Finding the worst poisonous point $S$ is computationally expensive for two reasons: we cannot use gradient descent to optimize over $S$ since the target function is implicit in $S$, and for each point $S$, a matrix inversion is required to find $\max_Q M(Q)$. Our analysis suggests that if $S$ is far enough from the data $D$, we can use $M(S)$ as a lower-bound estimate of $\max_Q M(Q)$.

### F.3 THE OPTIMAL QUERY FOR MEAN DISTANCE LOOD ON IMAGE DATASETS

As we concluded in the analysis in Section 4, only when the differing datapoint is far away (in some notion of distance) from the rest of the dataset is the optimal query (in mean distance LOOD) the differing datapoint. In Figure 11b, we successfully identified settings on a toy dataset where the differing point is not optimal for mean distance LOOD. We now investigate whether this also happens on an image dataset.

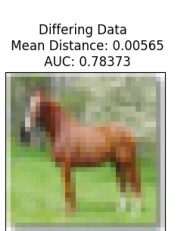 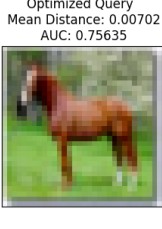 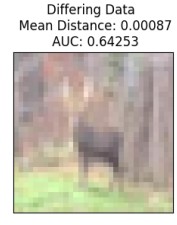 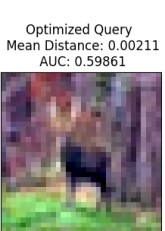 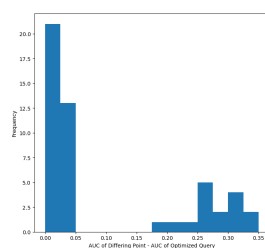

(a) Differing data and optimized query in run one

(b) Differing data and optimized query in run two

(c) AUC gap on $Q, S$ in 50 trials

Figure 13: Results for running single query optimization for Mean distance LOOD on class-balanced subset of CIFAR-10 dataset of size 1000 and RBF kernel. The optimized query has higher mean distance LOOD than the differing data record, despite having lower MIA AUC score. Across the 50 trials that we evaluated, more than 50% of trials result in an optimized query that has lower AUC score (by more than 0.025) than the differing data.

In Figure 13, we observe that the optimized record is indeed similar (but not identical) to the differing record for a majority fraction of cases. Therefore, also in image data the removal of a certain datapoint may influence the learned function most in a location that is not the removed datapoint itself. However, the optimized query for Mean distance LOOD still has much smaller MIA AUC score than the differing data. This again indicates that MSE optimization finds a suboptimal query for information leakage, despite achieving higher influence (mean distance).

## G DEFERRED PROOFS FOR SECTION 6

Our starting point is the following observation in prior work (Schoenholz et al., 2017; Hayou et al., 2019; 2021), which states that the NNGP kernel is sensitive to the choice of the activation function.

**Proposition G.1.** *Assume that the inputs are normalized, $\|x\| = 1$ and let $K_{NNGP}^L$ denote the NNGP kernel function of a an MLP of depth $L \geq 1$. Then, the for any two inputs, we have the following*
- *[Non-smooth activations] With ReLU: there exists a constant $\alpha > 0$ such that $|k_{NNGP}^L(x, x') - \alpha| = \Theta(L^{-2})$.*
- *[Smooth activations] With GeLU, ELU, Tanh: there exists a constant $\alpha > 0$ such that $|k_{NNGP}^L(x, x') - \alpha| = \Theta(L^{-1})$.*
*As a result, for the same depth $L$, one should expect that NNGP kernel matrix with ReLU activation is closer to a low rank constant matrix than with smooth activations such as GeLU, ELU, Tanh.*[12]

For proving Proposition 6.1, we will need the following lemma.

**Lemma G.2.** *Let $A$ and $B$ be two positive definite symmetric matrices. Then, the following holds*

$$\|A^{-1} - B^{-1}\|_2 \leq \max(\lambda_{min}(A)^{-1}, \lambda_{min}(B)^{-1})\|A - B\|_2,$$

*where $\lambda_{min}(A)$ is the smallest eigenvalue of A, and $\|.\|_2$ is the usual 2-norm for matrices.*

*Proof.* Define the function $f(t) = (tA + (1 - t)B)^{-1}$ for $t \in [0, 1]$. Standard results from linear algebra yield

$$f'(t) = -f(t)(A - B)f(t)$$

---

[12]The different activations are given by GeLU$(x) = x\Phi(x)$ where $\Phi$ is the c.d.f of a standard Gaussian variable, ELU$(x) = 1_{\{x \geq 1\}}x + 1_{\{x < 1\}}(e^x - 1)$, and Tanh$(x) = (e^x - e^{-x})(e^x + e^{-x})^{-1}$.

Therefore, we have

$$
\begin{aligned}
\|f(1) - f(0)\|_2 &= \|\int_0^1 f'(t)dt\|_2 \\
&\le \int_0^1 \|f'(t)\|_2 dt \\
&\le \int_0^1 \|f(t)\|_2^2 \|A - B\|_2 dt \\
&= \int_0^1 (\lambda_{min}(tA + (1-t)B))^{-2} \|A - B\|_2 dt.
\end{aligned}
$$

Using Weyl's inequality on the eigenvalues of the sum of two matrices, we have the following

$$
\begin{aligned}
\lambda_{min}(tA + (1-t)B) &\ge \lambda_{min}(tA) + \lambda_{min}((1-t)B) \\
&= t\lambda_{min}(A) + (1-t)\lambda_{min}(B) \\
&\ge \min(\lambda_{min}(A), \lambda_{min}(B)).
\end{aligned}
$$

Combining the two inequalities concludes the proof. $\square$

We are now ready to prove Proposition 6.1, as follows.

***Proposition 6.1.*** Let $D$ and $D' = D \cup S$ be an arbitrary pair of leave-one-out datasets, where $D$ contains $n$ records and $D'$ contains $n+1$ records. Define the function $h(\alpha) = \sup_{x,x' \in D'} |K(x, x') - \alpha|$. Let $\alpha_{min} = \text{argmin}_{\alpha \in \mathbb{R}^+} h(\alpha)$, and thus $h(\alpha_{min})$ quantifies how close the kernel matrix is to a low rank all-constant matrix. Assume that $\alpha_{min} > 0$. Then, it holds that

$$
\max_Q M(Q) \le C_{n,\alpha_{min},\zeta} \, n^{5/2} h(\alpha_{min}) + C_{\alpha,\zeta} \, n^{-1},
$$

where

$$
C_{n,\alpha_{min},\zeta} = \max\left( \frac{(\alpha_{min} + h(\alpha_{min}))\zeta}{\min(\alpha_{min}\sigma^2, |\alpha_{min}\sigma^2 - nh(\alpha_{min})|)}, \frac{(1+n^{-1})^{5/2}(\alpha_{min} + h(\alpha_{min}))\zeta}{\min(\alpha_{min}\sigma^2, |\alpha_{min}\sigma^2 - (n+1)h(\alpha_{min})|)} \right),
$$

and $C_{\alpha_{min},\zeta} = \zeta(1 + \alpha_{min}^{-1})$.

As a result, if $n^{5/2} h(\alpha_{min}) \ll 1$ and $n \gg 1$, then the maximum mean distance LOOD (i.e., maximum influence) satisfies $\max_Q M(Q) \ll 1$.

Similarly, there exist constants $A_n, B > 0$ such that

$$
\max_Q KL(f_{D,\sigma^2}(Q) \| f_{D',\sigma^2}(Q)) \le A_n h(\alpha_{min}) + B \, n^{-1}.
$$

*Proof.* Let $Q$ be a query point and $\alpha \in \mathbb{R}$. We have the following

$$
M(Q) = \frac{1}{2} \|K_{QD} M_D^{-1} y_D - K_{QD'} M_{D'}^{-1} y_{D'}\|,
$$

where $M_D = (K_{DD} + \sigma^2 I)$ and the same holds for $M_{D'}$. Let $U = \alpha U_n + \sigma^2 I$ and $U' = \alpha U_{n+1} + \sigma^2 I$, where $U_n$ is the $n \times n$ matrix having ones everywhere. With this notation, we have the following bound

$$
\begin{aligned}
2 \times M(Q) &\le \|K_{QD}(M_D^{-1} - U^{-1})y_D\| + \|K_{QD'}(M_{D'}^{-1} - (U')^{-1})y_{D'}\| \\
&\quad + \|K_{QD'}(U')^{-1}y_{D'} - K_{QD}U^{-1}y_D\|,
\end{aligned}
$$

Let us deal with first term. We have that

$$
\|K_{QD}(M_D^{-1} - U^{-1})y_D\| \le \|K_{QD}\| \|y_D\| \|M_D^{-1} - U^{-1}\| \le n^{1/2}(\alpha + h(\alpha))\|y_D\|_2 \|M_D^{-1} - U^{-1}\|.
$$

Using Lemma G.2, we have the following

$$\|M_D^{-1} - U^{-1}\| \leq \max(\lambda_{min}(A)^{-1}, \lambda_{min}(B)^{-1})\|M_D - U\|,$$

which implies that

$$\|K_{QD}(M_D^{-1} - U^{-1})y_D\| \leq \frac{n^{3/2}(\alpha + h(\alpha))\|y_D\|_2}{\min(\alpha\sigma^2, |\alpha\sigma^2 - nh(\alpha)|)} \times h(\alpha),$$

where we have used the inequality $|\lambda_{min}(A) - \lambda_{min}(B)| \leq \|A - B\|_2$ for any two symmetric positive matrices $A, B$.

Similarly, we have the following

$$\|K_{QD'}(M_{D'}^{-1} - (U')^{-1})y_D\| \leq \frac{(n+1)^{3/2}(\alpha + h(\alpha))\|y_{D'}\|_2}{\min(\alpha\sigma^2, |\alpha\sigma^2 - (n+1)h(\alpha)|)} \times h(\alpha).s$$

For the last term, we have

$$\begin{aligned}
\|K_{QD'}(U')^{-1}y_{D'} - K_{QD}U^{-1}y_D\| &\leq \|(K_{QD'} - \alpha e_{n+1}^\top)(U')^{-1}y_{D'}\| \\
&\quad + \|(K_{QD} - \alpha e_n^\top)U^{-1}y_D\| \\
&\quad + \|\alpha e_{n+1}^\top(U')^{-1}y_{D'} - \alpha e_n^\top U^{-1}y_D\|.
\end{aligned}$$

Moreover, we have

$$\|(K_{QD} - \alpha e_n^\top)U^{-1}y_D\| \leq \sigma^{-2}\|y_D\|\sqrt{n}h(\alpha),$$

and,

$$\|(K_{QD'} - \alpha e_{n+1}^\top)(U')^{-1}y_{D'}\| \leq \sigma^{-2}\|y_{D'}\|\sqrt{n+1}h(\alpha).$$

For the last term, we have

$$\begin{aligned}
\|\alpha e_{n+1}^\top(U')^{-1}y_{D'} - \alpha e_n^\top U^{-1}y_D\| &\leq \frac{\alpha}{|\alpha n + \sigma^2||\alpha(n+1) + \sigma^2|}\left|\sum_{i=1}^n y_{D_i}\right| + \frac{\alpha}{|\alpha(n+1) + \sigma^2|}|y_S| \\
&\leq (1 + \alpha^{-1})\zeta n^{-1}.
\end{aligned}$$

Combining all these inequalities yield the desired bound.

For the second upperbound, observe that $KL(f_{D,\sigma^2}(Q)\|f_{D',\sigma^2}(Q)) = \log\left(\frac{\Sigma'(Q)}{\Sigma(Q)}\right) + \frac{\Sigma(Q)}{2\Sigma'(Q)} + \frac{(\mu(Q) - \mu(Q'))^2}{\Sigma'(Q)} - \frac{1}{2}$ by the definition of KL LOOD in Definition C.3. Meanwhile, for sufficiently small $h(\alpha_{min})$, we have that

$$\Sigma(Q) = \alpha_{\min} - \alpha_{\min}^2 n(\sigma^2 + n\alpha_{\min})^{-1} + \mathcal{O}(h(\alpha_{\min})) = \alpha_{\min}\sigma^2(\sigma^2 + n\alpha_{\min})^{-1} + \mathcal{O}(h(\alpha_{\min})),$$

and a similar formula holds for $\Sigma'(Q)$ with $(n+1)$ instead of $n$. Therefore, we obtain

$$\frac{\Sigma(Q)}{\Sigma'(Q)} = 1 + \frac{\alpha_{\min}}{\sigma^2 + \alpha_{\min}n}.$$

Combining this result with the upperbound on the mean distance, the existence of $A_n$ and $B$ is straightforward. $\qquad\square$

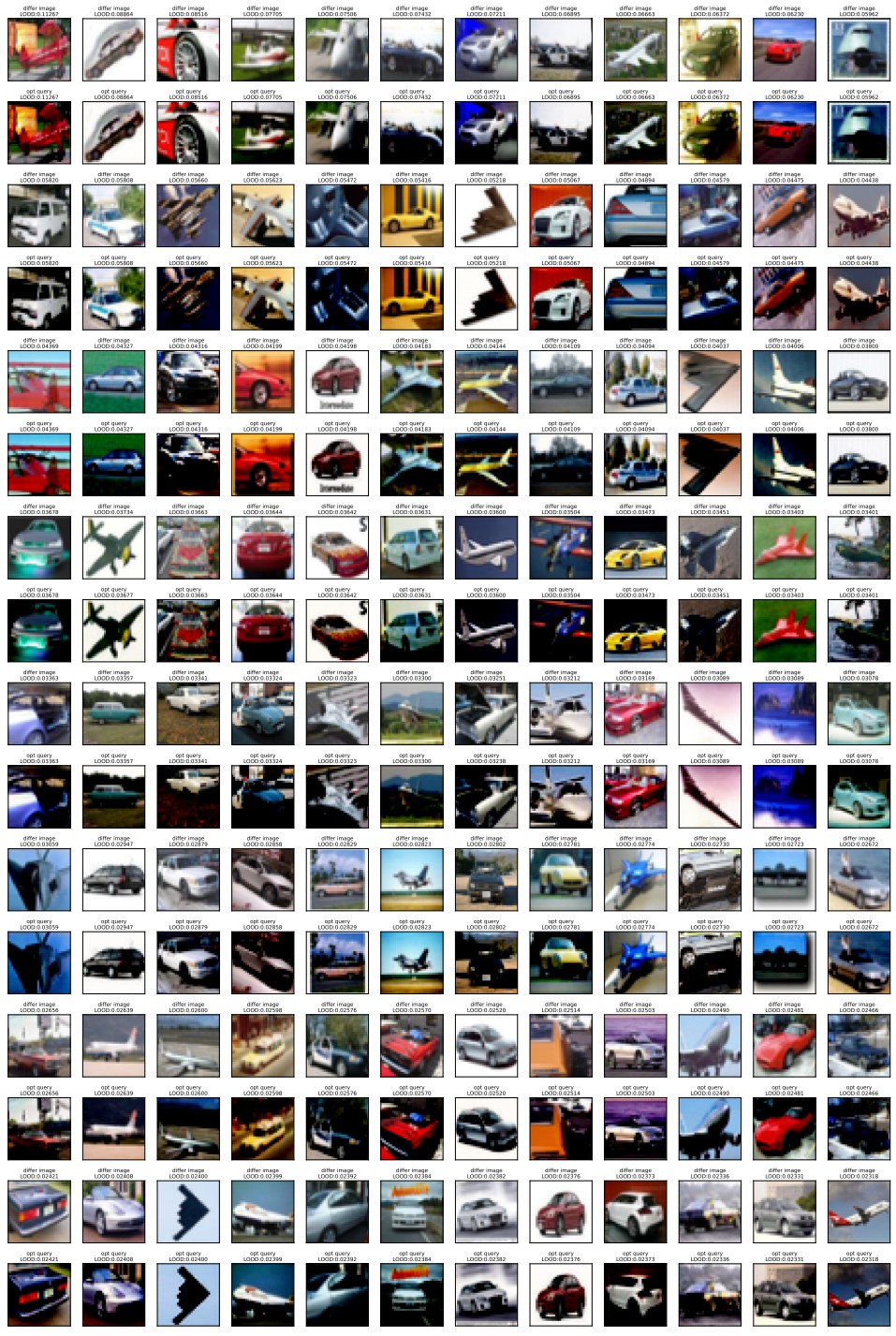

Figure 14: Visualization of 500 **randomly chosen** differing data and query optimized by LOOD (Part 1). We show differing data above the optimized query, and use their LOOD gap to sort the images.

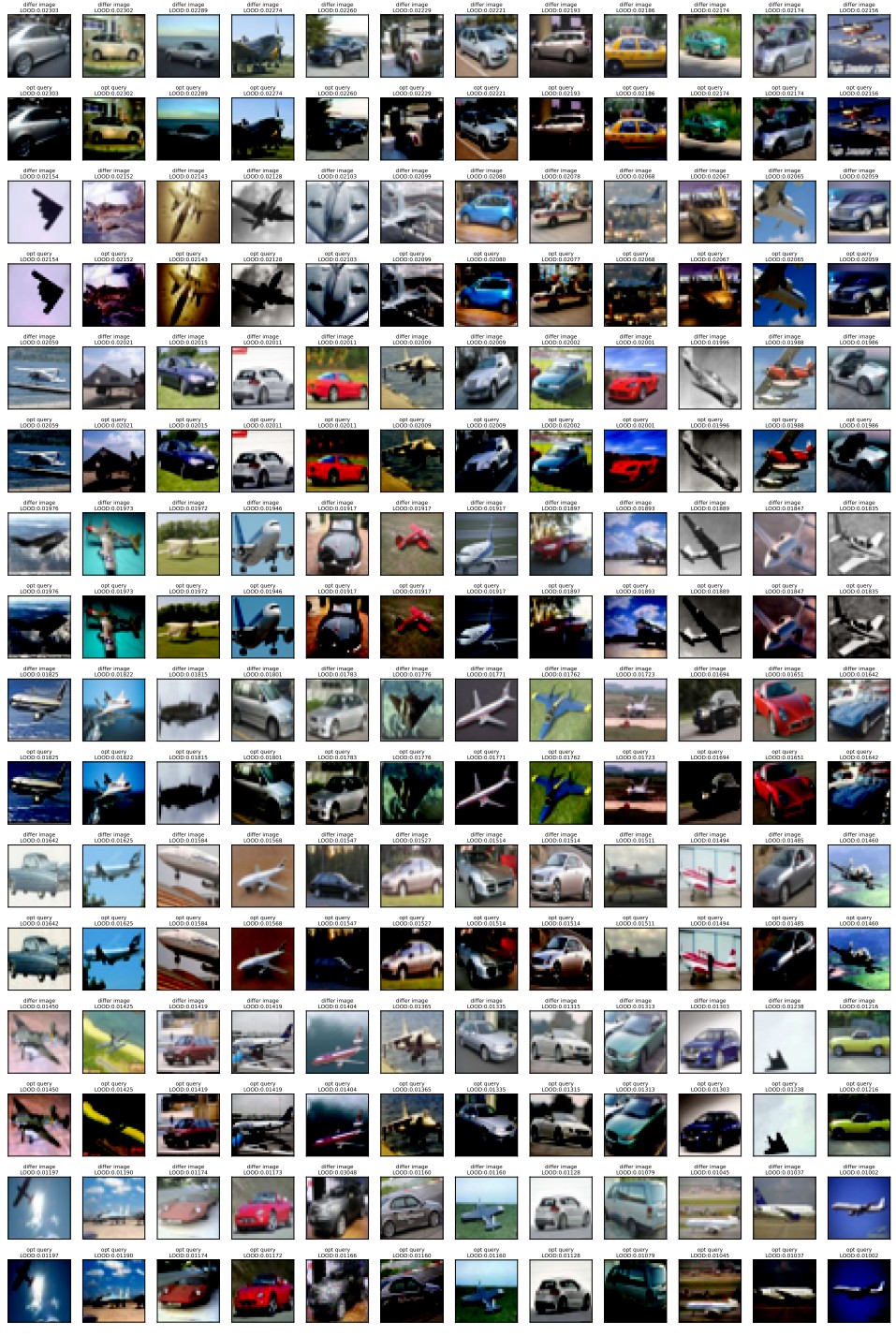

Figure 15: Visualization of 500 **randomly chosen** differing data and query optimized by LOOD (Part 2). We show differing data above the optimized query, and use their LOOD gap to sort the images.

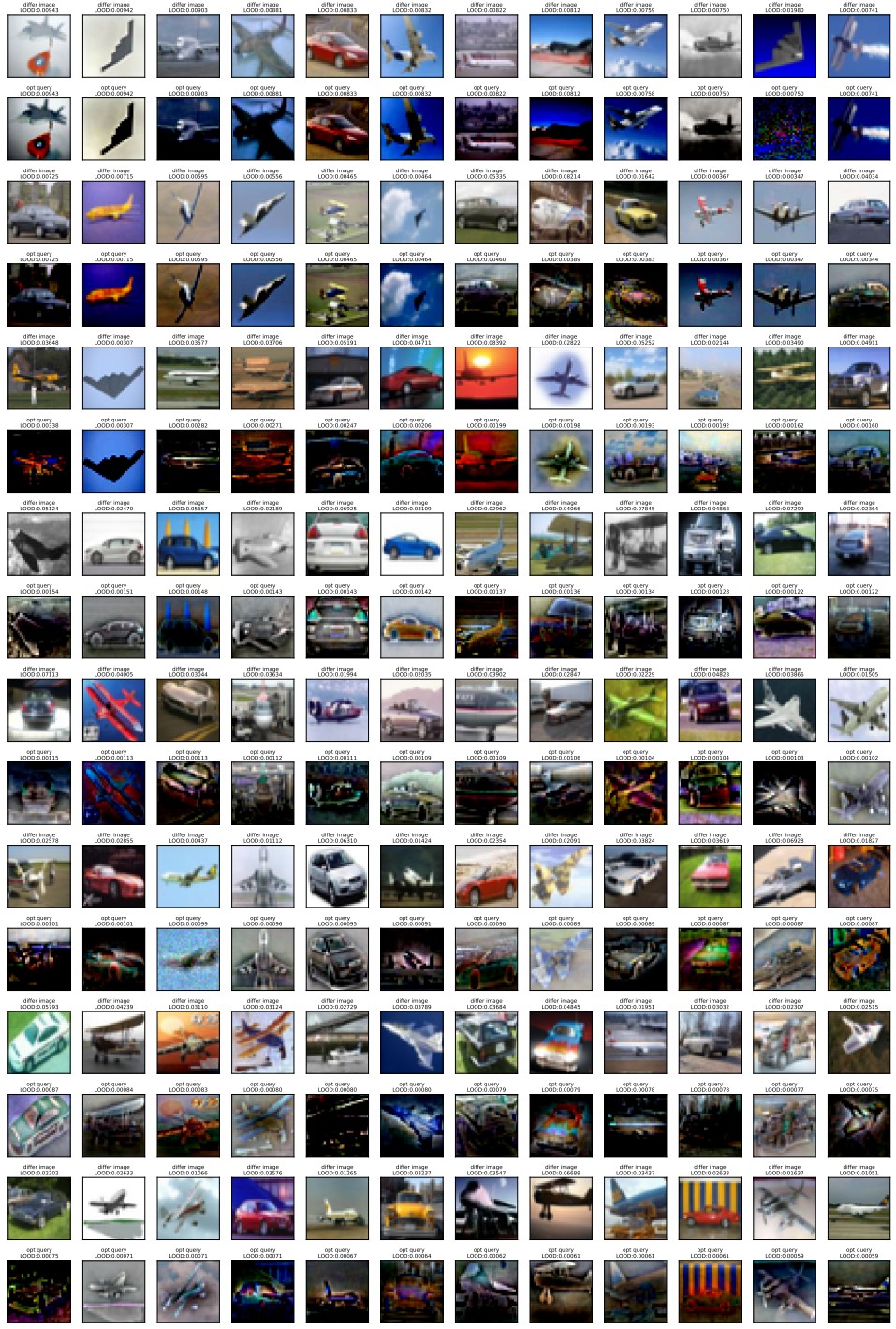

Figure 16: Visualization of 500 **randomly chosen** differing data and query optimized by LOOD (Part 3). We show differing data above the optimized query, and use their LOOD gap to sort the images.

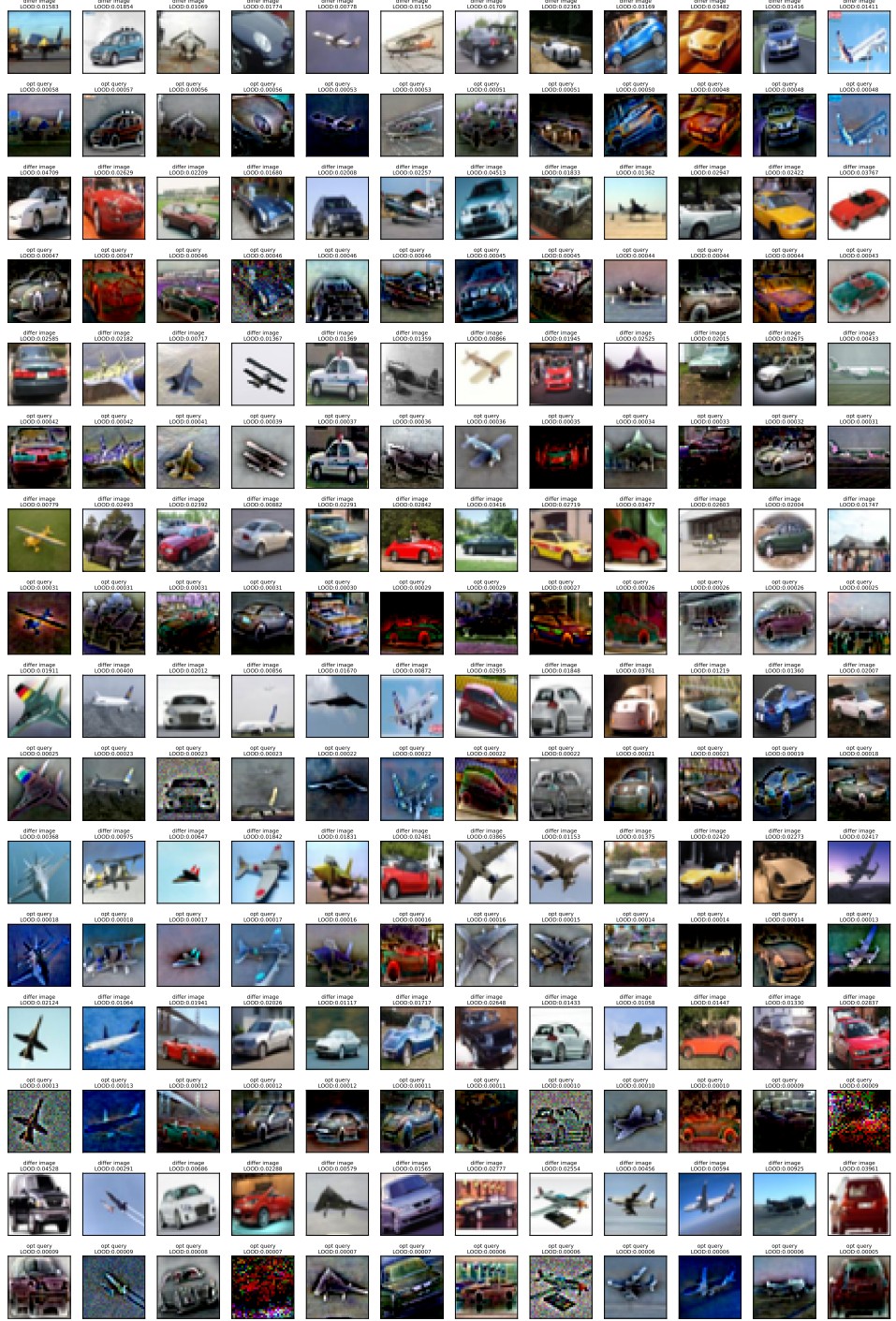

Figure 17: Visualization of 500 **randomly chosen** differing data and query optimized by LOOD (Part 4). We show differing data above the optimized query, and use their LOOD gap to sort the images.

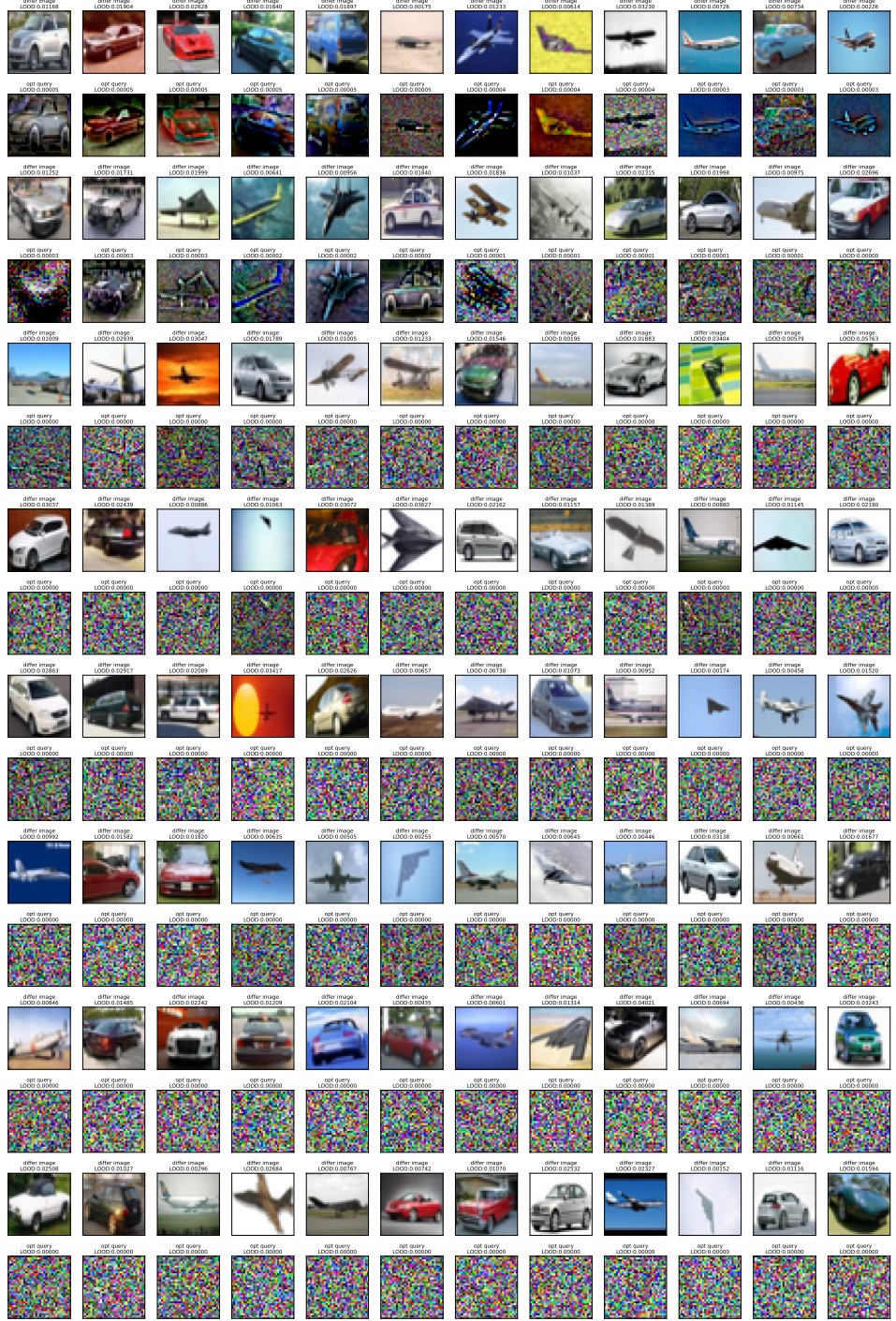

Figure 18: Visualization of 500 **randomly chosen** differing data and query optimized by LOOD (Part 5). We show differing data above the optimized query, and use their LOOD gap to sort the images.

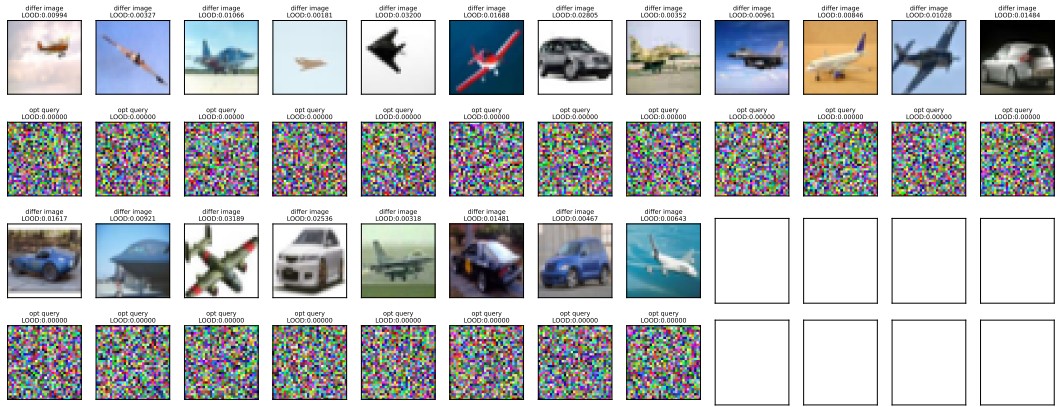

Figure 19: Visualization of 500 **randomly chosen** differing data and query optimized by LOOD (Part 6). We show differing data above the optimized query, and use their LOOD gap to sort the images.

