# OpenReview forum: "Leave-one-out Distinguishability in Machine Learning"
_ICLR.cc/2024/Conference — ICLR 2024 poster_

### Official Review · Reviewer_s1Kq · 2023-10-30

**Soundness:** 3 good
**Presentation:** 3 good
**Contribution:** 3 good
**Rating:** 5
**Confidence:** 4

**Summary:**

This paper introduces a comprehensive framework using Gaussian processes to analyze the influence of individual and groups of training data points on machine learning model predictions, with a focus on privacy implications. The authors propose Leave-One-Out Distinguishability (LOOD) as a metric to quantify information leakage and demonstrate its applicability in various settings, including Gaussian processes and deep neural networks (NNGPs). They also provide theoretical insights, especially on the impact of activation functions on information leakage, and validate their findings with extensive experiments.

**Strengths:**

The paper introduces a novel theoretical framework using Gaussian processes to analyze the influence of individual and groups of data points on machine learning model predictions. The authors creatively combine ideas from influence analysis, memorization, and information leakage, providing a comprehensive perspective on how training data affects model predictions. The application of LOOD (Leave-One-Out Distinguishability) as a measure of information leakage, and the exploration of its relationship with activation functions, add a unique angle to the existing body of work.

**Weaknesses:**

1.The authors have considered the fact that LOOD is a non-convex objective when analyzing global optimality using first-order information, acknowledging that analyzing global optima is quite challenging. However, if one solely utilizes the RBF kernel, as a nonlinear equation, many techniques from geometric analysis could be applied to examine the connections between global and local optima (such as considering the local properties of solutions to nonlinear equations using Sard's Lemma, etc.). Relying on experiments to complement the theoretical proof significantly undermines the credibility of the theory.

Similar issues are present in other sections of the paper as well. As an article that introduces a theoretical framework, the feasibility of this framework is supported by experimental evidence in many places. This approach raises doubts about the theoretical correctness of the framework, as it heavily relies on empirical validation rather than providing rigorous theoretical proofs throughout the paper.

2.The paper extensively analyzes the LOOD under RBF and NNGP kernels. While these are commonly used kernels, the generalizability of the results to other types of kernels or models is not clear. The paper could be strengthened by either extending the analysis to other kernels or by providing a clear justification for focusing on these specific kernels.

3.Given the definition of LOOD, using it to analyze and measure MIAs seems like a natural fit. However, the paper does not discuss whether LOOD can be used to measure the privacy capabilities against other potential types of attacks.

**Questions:**

1.Does the approach of modeling the randomness of models using Gaussian processes have general applicability beyond neural networks?

2.Given the definition of LOOD, using it to analyze and measure MIAs seems like a natural fit. However, for other potential attack methods, is there a way for LOOD to measure their privacy capabilities?

---

> ### Author Response · Authors · 2023-11-20
> **Response to reviewer s1Kq (1/2)**
>
> > W1: The authors have considered the fact that LOOD is a non-convex objective when analyzing global optimality using first-order information, acknowledging that analyzing global optima is quite challenging. However, if one solely utilizes the RBF kernel, as a nonlinear equation, many techniques from geometric analysis could be applied to examine the connections between global and local optima (such as considering the local properties of solutions to nonlinear equations using Sard's Lemma, etc.). Relying on experiments to complement the theoretical proof significantly undermines the credibility of the theory.
> >
> > As an article that introduces a theoretical framework, the feasibility of this framework is supported by experimental evidence in many places. This approach raises doubts about the theoretical correctness of the framework, as it heavily relies on empirical validation rather than providing rigorous theoretical proofs throughout the paper.
>
> We thank the reviewer for the suggestion of global optima analysis for RBF kernels. We are open to any more concrete pointers. However, our primary goal is to work with the more complex NNGP kernels due to their relation to NNs, for which we are not aware of any techniques that allow a theoretical analysis of the global optima of LOOD.
>
> The experiments for GPs are not done to complement proofs. Our analytical results for GPs are entirely independent of the numerical experiments, which we summarize as follows.
> - Stationary condition (Theorem 3.1 and Proposition 4.1) and second-order condition (Lemma D.4) of the LOOD optimization problem for a general class of kernel functions. Notably, the second-order condition (Lemma D.4) suffices to prove that mean distance LOOD is **locally maximal** when the query equals the differing data.
> - Complete theorems and proofs for how the activation choice affects the magnitude of LOOD in Proposition 5.1 and Proposition H.1.
>
> Instead, the numerical experiments for GPs are meant to (1) **confirm the analytical results** (e.g., Figure 2 (a)(b) and Figure 5 - NNGP curve); (2) **show the practical usefulness** of these analytical results (e.g., Figure 1, 4 and 14-19 shows how we can experimentally optimize LOOD to precisely reconstruct each training data, thus reflecting the usefulness of the stationary condition and second-order condition for LOOD); and (3) **compare** with prior **experimental** approaches for estimating information leakage, influence and memorization in quality (Figure 3) and efficiency (Footnote 4).
>
> Finally, we highlight that while introducing an analytical framework, one of our ultimate goals is to enable successful applications and insights for deep neural networks beyond theory. For this reason, we conduct a variety of experiments on deep neural network models to show how LOOD enables efficient and accurate estimates for influence, memorization, and information leakage (Figure 3) and how the analytical conclusions for LOOD generalize to neural networks (Figure 2 (c) regarding the stationarity and local optimality of LOOD and Figure 5 - NN curve regarding the effect of activation function choice on LOOD).
>
>
> > W2 & Q1: The paper extensively analyzes the LOOD under RBF and NNGP kernels. While these are commonly used kernels, the generalizability of the results to other types of kernels or models is not clear. The paper could be strengthened by either extending the analysis to other kernels or by providing a clear justification for focusing on these specific kernels.
> >
> > Does the approach of modeling the randomness of models using Gaussian processes have general applicability beyond neural networks?
>
> Our analysis for LOOD under GPs generally only requires weak regularity conditions that the kernel functions need to satisfy. For example, in Theorem 3.1 and Proposition 4.1 for proving the stationary condition, we only require the kernel function $K$ to be normalized (i.e., $K(x, x) = 1$ for any data $x$) and $\frac{\partial K(x, x)}{\partial x} = 0$ for any data $x$. Theorem 5.1 also applies to any bounded kernel functions. Consequently, our analytical results easily extend to kernel functions that satisfy these conditions, such as the general isotropic kernels [Proposition D.2] and correlation kernels [Proposition D.3].
>
> This paper focuses on the NNGP kernel function because we are interested in using LOOD for neural network models. To this end, the NNGP kernel is one of the most suitable kernel functions for our applications, as NNGP captures the infinite-width limit of neural networks. See Appendix B.2 for detailed discussions on how NNGPs connect to neural networks.

---

> ### Author Response · Authors · 2023-11-20
> **Response to reviewer s1Kq (2/2)**
>
> > W3 & Q2: Given the definition of LOOD, using it to analyze and measure MIAs seems like a natural fit. However, for other potential attack methods, is there a way for LOOD to measure their privacy capabilities?
>
> This is a good observation. The definition of LOOD is equivalent to distinguishability in a membership inference game, i.e., the prediction distributions of models trained on two neighboring datasets are indistinguishable. Such indistinguishability against membership inference is known to be fundamental, e.g., the existence of a strong attacker in the leave-one-out reconstruction game necessitates the existence of a strong membership inference attacker [a, Section II.D].
>
> This fundamental indistinguishability against membership inference is also the underlying intuition for differential privacy [b]. See our [response to reviewer rJTi](https://openreview.net/forum?id=9RNfX0ah0K&noteId=AmWjCVXOqL) for a detailed discussion on the connection between LOOD and DP. DP has been shown to imply indistinguishability against more advanced attribute inference games [c, d] or reconstruction games (see [a, Section VII], [e], and [f]). Therefore, similar to differential privacy, LOOD can also be used for semantically measuring the advantage of various inference attacks, such as reconstruction and attribute inference attacks. See [Section 5.2][e] for a concrete example of how KL divergence (also used in defining LOOD) translates to upper bound on the advantage of attribute inference attack.
>
> **References**
> - [a] Balle, B., Cherubin, G., & Hayes, J. (2022, May). Reconstructing training data with informed adversaries. In 2022 IEEE Symposium on Security and Privacy (SP) (pp. 1138-1156). IEEE.
> - [b] Dwork, C., McSherry, F., Nissim, K., & Smith, A. (2006). Calibrating noise to sensitivity in private data analysis. In Theory of Cryptography: Third Theory of Cryptography Conference, TCC 2006, New York, NY, USA, March 4-7, 2006. Proceedings 3 (pp. 265-284). Springer Berlin Heidelberg.
> - [c] Guo, C., Sablayrolles, A., & Sanjabi, M. Analyzing privacy leakage in machine learning via multiple hypothesis testing: A lesson from fano. In ICML 2023.
> - [d] Yeom, S., Giacomelli, I., Fredrikson, M., & Jha, S. (2018, July). Privacy risk in machine learning: Analyzing the connection to overfitting. In 2018 IEEE 31st Computer Security Foundations Symposium (CSF) (pp. 268-282). IEEE.
> - [e] Guo, C., Karrer, B., Chaudhuri, K., & van der Maaten, L. (2022, June). Bounding training data reconstruction in private (deep) learning. In International Conference on Machine Learning (pp. 8056-8071). PMLR.
> - [f] Hayes, J., Mahloujifar, S., & Balle, B. (2023). Bounding Training Data Reconstruction in DP-SGD. In NeurIPS 2023.

---

### Official Review · Reviewer_Azxn · 2023-11-01

**Soundness:** 3 good
**Presentation:** 3 good
**Contribution:** 3 good
**Rating:** 6
**Confidence:** 3

**Summary:**

This paper introduces a notion of algorithmic stability called *leave-one-out distinguishability*. For a learning algorithm that outputs a predictor, a pair of data sets $D$ and $D'$ that differ in a single record, and a specific query point $Q$, LOOD captures: how much does the algorithm's prediction on $Q$ change between $D$ and $D'$? This is a notion of local stability. The paper formalizes "change" through either KL divergence or average prediction, depending on the application. We also sometimes allow the data sets to differ in multiple points, or allow $Q$ to be a set of queries.

The paper aims to both unify existing work on memorization and influence and to answer questions about specific learning procedures. It contains a large number of both theoretical and experimental results. The theory results focus on Gaussian processes, while the experiments supplement these theorems and also show how the results can extend to neural networks.

After introducing the definition, we move to Section 3, which addresses the question of what query point maximizes LOOD. We see strong theoretical and experimental evidence pointing to the conclusion that, for GPs, the optimal query is the point that differs between $D$ and $D'$. (Nonconvexity makes establishing global optimality difficult.) We see an experiment with some evidence that the story is similar for neural networks.

Section 4 relates LOOD to existing notions of memorization and privacy. Most interesting to me is Figure 3a, which shows that data points tend to have similar susceptibility to membership inference attacks under NNGPs and NNs. We also see that GPs are vulnerable to data reconstruction attacks.

Finally, Section 5 explores how the activation function affects NNGPs and NNs. We get a theorem about the rank of NNGPs and complementary experiments on both classes of models.

**Strengths:**

This submission is full of ideas and deserves a treatment longer than nine pages. Despite that, the paper is relatively clear. The definition of LOOD closely builds on existing ideas in the literature, but uses them in new ways. The fact that we get clear results on GPs is nice, and I appreciate how we saw the results extended to NNs.

With so much content, I think it's a paper that will attract a large audience. I hope it gives them food for thought. I vote for acceptance.

**Weaknesses:**

The paper's density caused rushed discussions. In particular, I would have appreciated more "hand-holding" alongside the theorems and proofs. Similarly, I might prefer fewer experimental results with clearer explanations.

My largest critique of LOOD is that I feel it lacks a "killer application" which cleanly shows off its value. To elevate the paper, I would hope for a stronger answer to "what can we do now that we could not do before?" The paper gives answers about GPs, but I find these a bit underwhelming: I do not expect kernel methods to protect privacy. (Of course, people who do not work on privacy may have different expectations.) The results on NNs are interesting, but none of them are explored in enough detail to serve as a headline result.

**Questions:**

When having coffee with other researchers, which results from this paper are you most excited to discuss?

Which results in this paper will have the biggest impact on future research?

---

> ### Author Response · Authors · 2023-11-20
> **Response to reviewer Azxn (1/2)**
>
> > **Q1:** When having coffee with other researchers, which results from this paper are you most excited to discuss? **W1:** My largest critique of LOOD is that I feel it lacks a "killer application" which cleanly shows off its value... I would hope for a stronger answer to "what can we do now that we could not do before?"
>
> Understanding how each training data influences the prediction of the trained model on query data is a **canonical** problem to understand DNNs. It allows answering essential questions such as whether data X is used in training a model (Shokri et al., 2017), how robust the model is to removing data X from its training dataset (Koh & Liang, 2017; Koh et al., 2019; Pruthi et al., 2020), and how much does the model memorize individual training data (Feldman, 2020; Feldman & Zhang, 2020). Our driving motivation is to develop a framework that enhances the understanding of data influence on models. This framework aims to facilitate more efficient and interpretable answers to existing questions and to enable us to explore new questions about how a data point influences model predictions. Here are the interesting contributions of the paper:
> 1. We show how the estimations of information leakage, influence, and memorization are **intrinsically the same problem**. We propose one framework – leave-one-out distinguishability – that solves this problem accurately (Sections 4.1 and 4.3) and efficiently (Footnote 4).
> 2. For more advanced questions (e.g., what prediction leaks the most information, how the network activation choice affects leakage, and what is the joint influence of a group of training data), prior approaches [Shokri et al., 2017; Feldman & Zhang, 2020;Koh & Liang, 2017; Koh et al., 2019; Pruthi et al., 2020] are heavily based on experiments. Consequently, these prior approaches inevitably involve training many models under different training datasets or network architectures to analyze these advanced questions. Due to the experimental nature, conclusions derived from such approaches also tend to suffer from instability due to the large amount of uncontrollable experimental randomness. By contrast, LOOD is an analytical framework and allows one to efficiently and accurately answer such questions (Figure 2, Section 3, Section 5, Appendix E.2.) without the need for training any models – it only requires computing the NNGP kernel function using neural networks at initialization.
> 3. Lastly, the analytical nature of LOOD enables performing optimization to identify predictions that leak the most information about each training data record (Section 3 and Section 4.3), which enables exact reconstruction of each training data (Section 4.2). Such an optimization framework for influence sheds light on many interesting open problems, which prior experiment-based methods suffer from huge computation cost and experimental instability. We elaborate these new questions in response to Q2.
>
>
> > **Q2:** Which results in this paper will have the biggest impact on future research?
>
> We show that using the divergence metric (instead of the empirical mean distance) enables a precise measure of influence, to the extent that it allows very accurate reconstruction of training data. The new definition as well as the analytical framework can be used to answer (optimization) questions regarding what causes memorization, and what reflects the influence of memorized data.
>
> - LOOD offers an analytic estimation of influence, memorization, and information leakage that agrees well with experimental observations (Figure 3, Section 4.1 and 4.3).
>   - This is a training-free approach that is attractive in computation efficiency.
>   - Due to the analytical nature, LOOD complements prior experimental approaches (that suffers from large computation cost and numerical instability). LOOD allows accurate and efficient analysis for advanced questions, such as how the prediction data (Section 3) and the activation function (Section 5) choice affect information leakage for each training data. It is an exciting open problem to use LOOD to analytically study how other factors affect information leakage, such as the training data (distribution) and architecture choices.
> - We showcase how to optimize LOOD to identify the prediction that leaks the most information and how this enables exact reconstruction for each training data (for NNGPs).
>   - As NNGPs are closely connected to their corresponding neural networks, such results shed light on designing more powerful reconstruction attacks against deep neural networks by optimizing empirical approximations of the LOOD objective.
>   - Additionally, such results illustrate that the leakage in revealing the complete prediction distribution (rather than one sample) is so high that exact reconstruction is possible. This implies a trade-off between accurate uncertainty estimation and information leakage and calls for training algorithms that adequately consider (and optimize) these trade-offs.

---

> ### Author Response · Authors · 2023-11-20
> **Response to Reviewer Azxn (2/2)**
>
> > The paper gives answers about GPs, but I find these a bit underwhelming: I do not expect kernel methods to protect privacy. (Of course, people who do not work on privacy may have different expectations.)
>
> We would like to highlight one major difference between GPs and kernel methods: Gaussian process explicitly models the randomness in the algorithm, while kernel methods only models the posterior mean of the Gaussian process regression. Therefore, GPs are more suitable for modeling randomized training algorithms than kernel methods. For example, we are able to compute finite LOODs for predictions using GPs, which has privacy implications on various inference attacks, is connected to the fundamental membership inference game, and connects to differential privacy (see our [response to Reviewer s1Kq (2/2) [W3 & Q2]](https://openreview.net/forum?id=9RNfX0ah0K&noteId=tzkLvhOubI) for more detailed discussion).
>
> Finally, although we acknowledge that the correspondence between GPs and neural networks is only exact under **infinite** width, we would like to highlight that experimentally the agreement between LOOD and actual leave-one-out retraining is consistently high (Sections 4.1 and 4.3) for deep neural networks. This demonstrates the usefulness of using LOOD for NNGPs to model the influence and information leakage for NNs.

---

### Official Review · Reviewer_rJTi · 2023-11-03

**Soundness:** 3 good
**Presentation:** 3 good
**Contribution:** 3 good
**Rating:** 8
**Confidence:** 2

**Summary:**

This paper introduces a new way to measure or quantify the output distributions of a machine learning model on changing a few points in the input dataset, which they call, "leave-one-out distinguishability" or "LOOD". This is defined as the statistical distance (in this case, the KL-divergence) between the output distributions of the model on changing a few data points in the dataset. The main applications or advantages of introducing this notion of LOOD is that LOOD could be used to quantify (1) the memorization of data, (2) the leakage of information (via membership inference attacks), and (3) the influence of certain training data points on the model predictions. In this work, the applications of LOOD are illustrated via Gaussian processes, which they use to model the randomness of the machine learning models. They also show the effect of activation functions on LOOD. From their empirical results, they show LOOD as a good measure for all the above phenomena.

**Strengths:**

1. Their definition of LOOD captures the influence of data points in the trained machine learning models quite well, which they show for different phenomena, such as information leakage via membership inference attacks, and memorization.
2. Their experiments cover enough breadth, for example, by considering different kernels (like RBF and NNGP). So, the set of results seems comprehensive enough.

**Weaknesses:**

From my limited understanding of the subject, I can't find any significant weaknesses in this work.

**Questions:**

1. Have you thought about the connections between LOOD and differential privacy (DP)? As in, DP algorithms guarantee that a few data points cannot influence the output of the algorithm a lot, so will LOOD give any useful information about DP ML algorithms?

---

> ### Author Response · Authors · 2023-11-20
> **Response to reviewer rJTi**
>
> > Have you thought about the connections between LOOD and differential privacy (DP)? As in, DP algorithms guarantee that a few data points cannot influence the output of the algorithm a lot, so will LOOD give any useful information about DP ML algorithms?
>
> **Regarding the connection between DP and LOOD**
>
> At the core, DP and LOOD are closely related. They both quantify the privacy risk of including a data point in the training set by evaluating the divergence in the algorithm's output distribution on neighboring datasets. LOOD calculates the privacy loss for individual samples in a black-box setting, whereas DP, in its current application, determines the maximum white-box privacy loss across all possible datasets.
>
> In the black-box setting, the threat model assumes that an adversary can access only the predictions made by a model, related to the scenario with API access to models. Conversely, DP addresses privacy loss under a stronger white-box threat model, where the adversary is assumed to have full access to the parameters of the trained model. The guarantee provided by DP certainly applies to the black-box setting too, but its primary analysis is done for the stronger adversary scenario in the white-box setting.
>
> There is a difference between the notions of divergence used in LOOD versus DP. LOOD is defined by the KL divergence between distributions, while standard DP is equivalent to bounded hockey-stick divergence between distributions. The closest DP definition to LOOD is the Rényi DP definition (as Rényi divergence with order $\alpha \rightarrow 1$ is equivalent to KL divergence).
>
> **Will LOOD give any useful information about DP ML algorithms?**
>
> DP guarantees that the prediction distributions cannot change beyond a certain bound under the change of one training data. However, such a guarantee reflects the worst-case scenario and the same bound holds for **any** possible training data and query. Thus, DP's bound doesn't discern the privacy risk of specific data points in certain training sets and queries. DP doesn't indicate if some predictions pose higher information leakage risks. LOOD, however, helps explore how information leakage varies with different predictions, which prediction leads to the most leakage, and how leakage evolves with more predictions, as detailed in Figure 2, Section 3, and Appendix E.1.

---

> > ### Comment · Reviewer_rJTi · 2023-11-20
> > **Thanks**
> >
> > Thanks! Updated my score.

---

### Official Review · Reviewer_mmKM · 2023-12-08

**Soundness:** 3 good
**Presentation:** 2 fair
**Contribution:** 3 good
**Rating:** 6
**Confidence:** 3

**Summary:**

The paper proposes a framework called LOOD that measures how a model's output on some samples Q changes when a set of samples S are added to the training set. It uses GP to model the model randomness and looks at the KL divergence or mean predictions between the distribution of the output on Q given datasets D and D' that differ in S. The Q that is affected by S the most is mostly S itself.
The paper then shows that LOOD can be applied to measure information leakage, data reconstruction, and influence. It can also be used to explain the effect of different activation functions.

**Strengths:**

The proposed framework is good at capturing many different aspects of model memorization and is computationally efficient.
The empirical evaluations are pretty throughout and interesting.

**Weaknesses:**

The presentation might be improved to make the contributions easier to see and the analyses easier to follow. For example, I was a bit confused about the purpose when I first saw Section 3 on "OPTIMIZING LOOD TO IDENTIFY THE MOST INFLUENCED POINT". Maybe a more detailed explanation on why we want to do so would help (e.g. how would knowing the most influenced point benefit us in measuring memorization / leakage etc).
It is also a bit unclear what LOOD would enable us to do that cannot be done with previous method. It seems to me that one big advantage of LOOD is the computation efficiency. If so, I think the authors can consider adding more detail comparison to highlight that.

**Questions:**

Maybe I'm missing some important point but why is some experiment done on training data that consists of two classes only (car & airplane)?

---

### Meta-Review · Area_Chair_ek9i · 2023-12-07

**Metareview:**

The paper got mixed reviews leaning on the positive side. While the reviewers agreed that the LOOD provides a new view point of understanding memorization/leakage, there is no killer app. After reading the paper, I landed up in a similar conclusion. Also, there were presentation issues raised by the reviewers (including the newest review we obtained from a reviewer who was delayed.) We would recommend the authors to focus on the narrative, and strive to demonstrate a clear application beyond Gaussian process.

**Justification For Why Not Higher Score:**

In my opinion, the application of the theoretical results is a bit too restricted. The paper might benefit a lot if the narrative is changed to make it a more empirical paper, with theoretical analysis is solely used to motivate the empirics.

**Justification For Why Not Lower Score:**

The reviews were all on the positive side, and the question is important.

---

### Decision · Program_Chairs · 2024-01-16

Accept (poster)